# CaTS-Bench: Can Language Models Describe Numeric Time Series?

## Abstract

Time series captioning, the task of describing numeric time series in natural language, requires numeric reasoning, trend interpretation, and contextual understanding. Existing benchmarks, however, often rely on synthetic data or overly simplistic captions, and typically neglect metadata and visual representations. To close this gap, we introduce **CaTS-Bench**, the first large-scale, real-world benchmark for **C**ontext-**a**ware **T**ime **S**eries captioning. CaTS-Bench is derived from 11 diverse datasets reframed as captioning and Q&A tasks, comprising roughly $465k$ training and $105k$ test timestamps. Each sample includes a numeric series segment, contextual metadata, a line-chart image, and a caption. A key contribution of this work is the scalable pipeline used to generate reference captions: while most references are produced by an oracle LLM and verified through factual checks, human indistinguishability studies, and diversity analyses, we also provide a human-revisited subset of 579 test captions, refined from LLM outputs to ensure accuracy and human-like style. Beyond captioning, CaTS-Bench offers 460 multiple-choice questions targeting deeper aspects of time series reasoning. We further propose new tailored evaluation metrics and benchmark leading VLMs, highlighting both their strengths and persistent limitations. Together, these contributions establish CaTS-Bench and its captioning pipeline as a reliable and extensible foundation for future research at the intersection of time series analysis and foundation models.

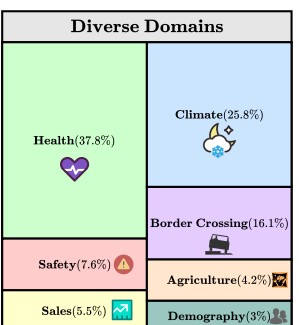
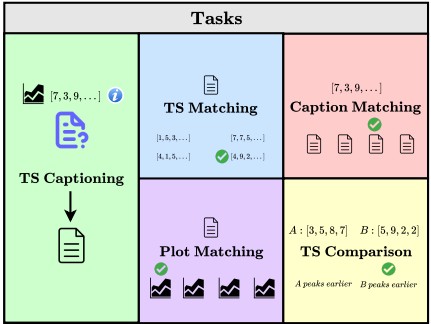
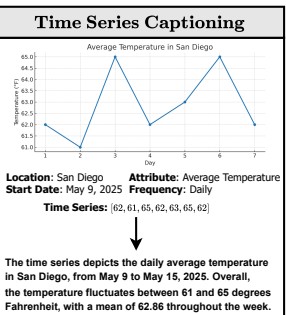

Figure 1: Overview of CaTS-Bench. It features diverse domains, provides training and benchmark data, and formulates five challenging tasks, with time series captioning as the primary one.

## 1 Introduction

Effective interpretation of time series data is a cornerstone of decision-making in domains ranging from financial markets and healthcare monitoring to climate analysis and industrial automation. Yet, distilling raw numeric sequences into concise, human-readable summaries remains a labor-intensive task, requiring domain expertise, statistical know-how, and careful visualization. Automating this process through *time series captioning* (TSC) not only accelerates insight discovery but also democratizes access to complex temporal analytics, enabling non-experts to ask natural-language questions and receive meaningful explanations without writing code or inspecting raw charts.

Large language models (LLMs) and vision-language models (VLMs) have demonstrated remarkable prowess in text generation and visual reasoning, respectively. However, when applied to time series, they reveal critical deficiencies: LLMs exhibit well-documented limitations in precise numeric extrapolation, temporal continuity, and uncertainty quantification (Tang et al., 2025; Merrill et al., 2024; Tan et al., 2024; Cao & Wang, 2024). While VLMs have shown promise in visual pattern recognition tasks such as trend and anomaly detection from plots (Zhou & Yu, 2025), their capacity for fine-grained numeric time series reasoning remains largely underexplored. These limitations underscore a broader challenge: existing evaluation resources fail to reflect the complexity of real-world temporal signals, leaving model improvements unguided by the demands of true data-driven applications.

In response, the community has proposed Time Series Captioning (TSC) as a more natural task for foundation models, leveraging their generative and reasoning capabilities to narrate trends, anomalies, and context in prose (Trabelsi et al., 2025; Jhamtani & Berg-Kirkpatrick, 2021). However, current benchmarks remain narrow, often synthetic or restricted to simple trend labels, and exclude rich metadata or visual modalities. Consequently, progress in model architecture, pretraining, or finetuning cannot be measured against challenges that mirror real deployment scenarios, slowing adoption in high-stakes sectors where accurate temporal interpretation is essential.

To fill this gap, we introduce **CaTS-Bench**, the first large-scale, multimodal benchmark explicitly designed for *context-aware* time series captioning and reasoning. We define "context-aware" to mean that captions are informed by both the metadata (units, domain labels, dates, region, etc.) and visual cues that provide semantic and numeric grounding. By mining 11 real-world datasets across various domains, CaTS-Bench provides $20k$ triplet samples drawn from $570k$ time steps of curated data, each paired with (1) rich metadata containing contextual information, units, and domain-specific cues (Dong et al., 2024; Wang et al., 2024); (2) a corresponding line plot image, enabling the use of VLMs (Chen et al., 2024a; Zhou & Yu, 2025); and (3) a reference caption produced by a scalable oracle-based pipeline and validated through factual checks, human indistinguishability studies, and diversity analyses. To further strengthen reliability, we additionally release a *human-revisited subset* of test captions: sampled from multiple LLM candidates and carefully edited by the authors to remove inaccuracies, speculative claims, and linguistic repetitions. This subset complements the larger benchmark with high-fidelity, human-styled references. Beyond captioning, CaTS-Bench also includes 460 challenging multiple-choice questions spanning time series matching, caption matching, plot matching, and comparative reasoning, designed to expose models' blind spots in numeric precision and multimodal alignment. All data samples are made available here.

We further propose new evaluation metrics tailored to time series captioning that move past generic N-gram overlap to reward numeric fidelity and coverage. Our comprehensive experiments on leading VLMs reveal that, in both zero-shot and finetuned settings, models can produce fluent text but fail to reliably capture quantitative details without specialized adaptation. A key finding is that VLMs fail to effectively leverage the visual cues provided for time series captioning, pointing to a significant limitation in current multimodal architectures. Our analysis identifies clear room for improvement, such as better leveraging visual cues, enhancing multimodal alignment, and incorporating dedicated numeric reasoning modules. These findings pave the way for a new generation of foundation models capable of translating complex temporal data into actionable narratives.

In summary, the contributions of this paper are:

1. **Scalable Captioning Pipeline**: A reproducible pipeline for generating high-quality time series captions. It anchors LLM outputs in factual metadata, validates them through factual checks, human indistinguishability studies, and diversity analyses, and is extensible to new datasets.

2. **CaTS-Bench**: A multimodal, context-aware benchmark for time series captioning and reasoning, featuring time series segments, rich metadata, visual plots, and factually grounded captions. Most references are LLM-generated via the pipeline, while a curated subset of human-revisited test captions ensures high-fidelity, human-styled references alongside the larger benchmark.

3. **Diagnostic Q&A Suite**: Four multiple-choice tasks designed to isolate capabilities in series matching, caption grounding, visual reasoning, and comparative analysis.

4. **Comprehensive Evaluation**: Zero-shot and finetuned assessments of state-of-the-art VLMs, revealing strengths, failure modes, and clear directions to advance time series understanding.

## 2 RELATED WORK

LLMs are increasingly being repurposed for time series analysis (Zhang et al., 2024; Liu et al., 2024a), with early efforts primarily focused on forecasting. These approaches span prompt engineering (Liu et al., 2024a; Chatzigeorgakidis et al., 2024), modality alignment (Liu et al., 2024b; Sun et al., 2023; Liu et al., 2024c; Pan et al., 2024), discretization (Ansari et al., 2024; Jin et al., 2024), and specialized finetuning (Zhou et al., 2023; Chang et al., 2023). Such studies highlight that LLMs pretrained on text can reason over temporal data, but subsequent work also shows consistent weaknesses in handling long-range dependencies, numeric precision, and structured reasoning, particularly in forecasting and anomaly detection (Tang et al., 2025; Merrill et al., 2024; Tan et al., 2024; Cao & Wang, 2024; Zeng et al., 2023).

Table 1: Comparison of TSC benchmarks.

| Dataset | # Timesteps | Modality | Sources | Metadata | Captions | TSC | Q&A |
|---|---|---|---|---|---|---|---|
| TADACap (Fons et al., 2024) | N/A | Visual | 4 | Minimal | Patterns Only | ✓ | ✗ |
| TRUCE (Jhamtani & Berg-Kirkpatrick, 2021) | 34k | Numeric | 2 | ✗ | Patterns Only | ✓ | ✗ |
| TACO (Dohi et al., 2025) | 2.46b | Numeric | 8 | ✗ | Expressive | ✓ | ✗ |
| **CaTS-Bench** | 570k | Numeric + Text + Visual | 11 | Rich | Expressive | ✓ | ✓ |

Building on these foundations, researchers have explored Time Series Captioning (TSC), a task more aligned with the generative strengths of language models. TSLM (Trabelsi et al., 2025) introduces an encoder–decoder trained on synthetic cross-modal data; TADACap (Fons et al., 2024) retrieves domain-aware captions for visualized time series; TRUCE (Jhamtani & Berg-Kirkpatrick, 2021) employs a truth-conditional framework to validate simple trend patterns; and TACO (Dohi et al., 2025) scales up caption corpora using LLM-based synthetic generation. While each provides valuable insights, they remain limited in scope: TADACap and TRUCE are domain-specific and pattern-oriented, while TACO's reliance on templates restricts contextual richness (See Table 1).

Beyond these, standard time-series archives such as UCR (Chen et al., 2015), UEA (Bagnall et al., 2018), and Monash (Godahewa et al., 2021) support classification and forecasting but not generative captioning. Similarly, benchmarks like PISA (Xue & Salim, 2023) target prompt-based forecasting, omitting metadata entirely. Recent evidence shows that incorporating auxiliary modalities (metadata, domain context, or visual renderings) can significantly improve both interpretability and predictive performance (Zhou & Yu, 2025; Dong et al., 2024; Chen et al., 2024a; Wang et al., 2024; Kim et al., 2024; Williams et al., 2024; Liu et al., 2025; Tang et al., 2023). Yet no benchmark to date integrates large-scale numeric series, expressive captions, rich metadata, and multimodal grounding. CaTS-Bench fills this gap by offering the first benchmark that unifies numeric time series, metadata, and visuals with both expressive captions and Q&A tasks for systematic evaluation for TSC.

## 3 CaTS-BENCH

In this section, we illustrate the entire data curation pipeline and the design of the benchmark tasks. While examples generated from this pipeline can be directly used for TSC evaluation, we further enrich the scope of CaTS-Bench by providing an additional suite of Q&A tasks constructed from the same data, enabling a more fine-grained examination of time series and caption reasoning abilities.

### 3.1 DATA CURATION

We build **CaTS-Bench**, a comprehensive benchmark curated from 11 diverse real-world source datasets spanning domains: climate (Jha, 2023; Ritchie, 2021), safety (of Los Angeles, n.d.; of Public Health, n.d.), USA border crossing (U.S. Department of Transportation, n.d.), demography (Aziz, 1985), health (European Centre for Disease Prevention and Control, 2024; Food and Agriculture Organization of the United Nations, 2024), sales (Hassan, 2020; Chen, 2015), and agriculture (USDA Economic Research Service, 2024). See Appendix B for more details on the source datasets. The overall data pipeline is shown in Figure 2. Each source dataset provides a full-length time series per entity (e.g., country, city, product), and to generate samples, we apply a random window cropping strategy. For each dataset, we define a valid range of window lengths and randomly select a size for each crop; see Appendix C for our range calculation. The number of windows sampled from a dataset

depends on its total time steps, ensuring fair representation. The domain-specific number and lengths of the time series windows are illustrated in Table 2. Each time series window is augmented with a **metadata JSON** file with contextual information (domain, location, start time, etc.), a **line plot image** with randomized visual style (color, width, figure size), a **ground truth caption** produced by querying an oracle LLM (`Gemini 2.0 Flash`) with a structured prompt that includes: (i) the serialized numeric values of the cropped segment and (ii) metadata enriched with numericly grounded information, including both the historical and sample-specific mean, standard deviation, minimum, and maximum. An example of the prompt is available in Appendix N.1.

We emphasize that time series captioning lacks inherent ground truth at the level of a single canonical description: multiple valid ways exist to describe the same series depending on focus and phrasing. To provide consistent references at scale, our primary captions are generated by an oracle LLM, but anchored strictly in the underlying data. The oracle receives full contextual metadata (not available at evaluation time) and is instructed not to include any external knowledge, ensur-

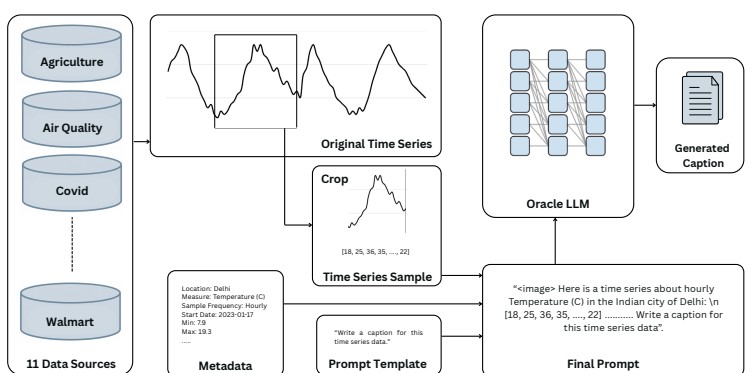

Figure 2: Overview of the CaTS-Bench semi-synthetic data generation pipeline. A time series window is cropped, metadata is attached, and an oracle LLM generates a reference caption. See Appendix L for examples and Appendix H for the quality verification protocol.

ing captions remain factual and context-grounded. This design makes captions a practical proxy for evaluation and challenges models to reason from multimodal inputs rather than mimic the oracle. Furthermore, we randomize time series window sizes and plot styles to prevent overfitting and better reflect real-world variability in length and visualization styles.

Table 2: Dataset outline by domain. AQ: Air Quality, Border: Border Crossing, Demo: Demography, Injury: Road Injuries, Calories: Calories Consumption, Agri: Agriculture

| Metric | Overall | AQ | Border | Crime | Demo | Injury | COVID | $CO_2$ | Calories | Walmart | Retail | Agri |
|---|---|---|---|---|---|---|---|---|---|---|---|---|
| # Source Time Steps | 287M | 286M | 397k | 38k | 14k | 37k | 720k | 34k | 234k | 6k | 7k | 49k |
| # Samples Generated | 20k | 4.4k | 3.2k | 764 | 598 | 756 | 5.5k | 732 | 2.1k | 544 | 551 | 835 |
| # Train Samples | 16k | 3.5k | 2.6k | 611 | 478 | 604 | 4.4k | 585 | 1.7k | 435 | 440 | 668 |
| Avg. Sample Length | 29.1 | 65.3 | 21.2 | 76.8 | 11.6 | 5.9 | 75.8 | 9.5 | 12.2 | 12.2 | 22.4 | 7.3 |
| # Test Samples | 4k | 886 | 646 | 153 | 120 | 152 | 1.1k | 147 | 422 | 109 | 111 | 167 |
| Avg. Sample Length | 26.1 | 66.0 | 21.2 | 76.9 | 5.0 | 3.6 | 73.0 | 8.7 | 5.5 | 11.8 | 8.1 | 7.5 |
| # Human-revisited Samples | 579 | 0 | 0 | 153 | 120 | 0 | 0 | 0 | 0 | 109 | 0 | 167 |
| Avg. Sample Length | 25.7 | - | - | 76.9 | 5.0 | - | - | - | - | 11.8 | - | 7.5 |

To prevent information leakage, we partition each source dataset temporally before generating the samples. Specifically, the first $80\%$ is used for generating training samples, whereas the last $20\%$ is reserved exclusively for generating test samples. Random window cropping is applied separately to the training and test partitions. This strategy ensures that the model is evaluated on future, unseen data relative to the training set. The actual benchmark samples consist of the test split resulting from this process. We leave the training split of the data for optional training. Our final semi-synthetic dataset version contains $20k$ examples, split into roughly $16k$ training samples and $4k$ test samples. Detailed statistics and source of our data are reported in Table 2.

**Human-Revisited Subset.** We also release a curated subset of test captions that have been revisited by humans. These captions were first sampled from multiple LLM candidates (`Gemini 2.0 Flash`, `GPT-4o`, `Gemma 27B`, and `Llama 90B`) using the above data pipeline, and then carefully refined by the authors to eliminate factual errors, speculative statements, and redundant phrasing.

Drawn from the domains of agriculture, crime, demography, and Walmart sales, this subset provides high-fidelity, human-styled references that complement the larger benchmark.

## 3.2 QUALITY VALIDATION OF SEMI-SYNTHETIC CAPTIONS

To ensure the quality of CaTS-Bench, we conducted a series of comprehensive verification studies addressing the core concern of semi-synthetic data: whether captions generated by the oracle model (Gemini 2.0 Flash) are factual, unbiased, and linguistically diverse. These analyses demonstrate that semi-synthetic captions in CaTS-Bench provide high-quality references and stable benchmarks for TSC, and thus are a sufficient proxy for human-written descriptions in practical scenarios. We verified caption quality through three complementary studies below (with full details in Appendix H).

**Manual Validation.** We manually checked $\sim 2.9k$ captions (72.5% of the semi-synthetic test benchmark) across statistical claims (min, max, mean, STD) and trend descriptors (up/downward, stable, fluctuating). Accuracy exceeded 98.6% on average across all categories (Table 9) which confirms that captions faithfully reflect underlying series properties.

**Human Detectability Study.** In a blind test with 35 participants, subjects attempted to distinguish our captions from those written by humans. Accuracy was near random at 41.1%, suggesting that our captions are indistinguishable from human-authored ones and no evidence of oracle-specific bias.

**Diversity and Bias Analysis.** Captions consistently drew from a wide variety of statistical and temporal descriptors (Table 12), and embedding-based similarity analysis across nine embedding models revealed minimal template reliance. Pairs of captions that were almost semantically identical, measured as embedding cosine similarity $> 0.95$, were rare, averaging 2.3% of occurrences (Table 13). Comparisons with human captions ( H.4.4) indicate that Gemini's outputs are stylistically intermixed with human text, while N-gram analysis (H.4.2) confirms high lexical diversity.

## 3.3 TIME SERIES CAPTIONING

TSC requires generating a detailed, coherent narrative that highlights the key characteristics of a given time series. During evaluation, each model is presented with a standardized multi-part prompt that combines four elements: the **numeric series** itself, embedded as raw time-indexed values (e.g., [25.3, 26.1, 26.8, ...]); **contextual metadata** such as measurement units, data source, sampling interval, and domain tags (e.g., "Hourly temperature readings from Rome, May 2000"), which excludes explicit statistics like mean or maximum since the model must infer them; a **visual input** in the form of a line-plot image that allows vision-language models to ground their descriptions in visual trend cues; and a fixed-format **instruction template** containing the directive for caption generation (see Appendix N.2). By standardizing this multi-part prompt, we evaluate models on their ability to recognize numeric trends (e.g., rising or falling segments, peaks, and troughs), integrate metadata cues, and utilize visual features to produce context-aware captions.

## 3.4 Q&A MULTIPLE-CHOICE TASKS

We introduce a suite of multiple-choice Q&A tasks designed to probe different reasoning skills in time series understanding. All tasks are automatically derived from the same source data used for captioning, with questions generated from task-specific, fixed templates (see Appendix J.1 for examples). To increase difficulty, an initial pool of $4k$ questions per type was filtered by removing those correctly answered by Qwen 2.5 Omni. Appendix J.2 shows that this filtering produces genuinely harder questions, rather than reflecting Qwen-specific weaknesses only. Ambiguous Time Series Matching questions were manually checked to ensure a single correct answer. From the remaining $7k$ challenging questions, a random subset of 460 was sampled as the final test set, including 100 each for time series matching, caption matching, and plot matching, and 40 each for amplitude, peak, mean, and variance comparison tasks. Question types are described below.

**Time Series Matching.** Given a caption, the model must retrieve the correct time series from distractor candidates created via shuffling, temporal reversal, and Gaussian noise. These perturbations prevent simple numeric lookup and require alignment with both values and trends (see J.3 for details).

**Caption Matching.** Given a time series, the model must select the correct caption from distractors composed of random captions and perturbed variants of the ground truth (see Appendix N.5, N.6). This isolates caption understanding from free-form generation.

**Plot Matching.** Given a caption and its numeric series, the model must select the correct line plot from the candidates, testing visual grounding and the ability to link language with visual patterns.

**Time Series Comparison.** Given two time series, select the correct comparative statement from a pair of options (e.g., "Series A peaks earlier than Series B" or "Series B has a higher volatility than Series A"). This task challenges models to perform temporal and statistical comparison, a setting where many language models currently struggle (Merrill et al., 2024).

### 3.5 EVALUATION METRICS

To comprehensively evaluate model-generated captions against the ground truth in TSC, we employ a diverse set of metrics that target linguistic quality, statistical inference, and numeric fidelity. For Q&A, we adopt accuracy as the evaluation metric, as each question is designed to have a single correct answer. Below, we describe each metric used for TSC in our evaluation framework.

**Standard Linguistic Metrics.** We assess caption similarity using standard NLP metrics, including DEBERTA SCORE (Zhang* et al., 2020), BLEU (Papineni et al., 2002), ROUGE-L (Chin-Yew, 2004), METEOR (Banerjee & Lavie, 2005), and SIMCSE (Gao et al., 2021; Liu et al., 2019). Together, these metrics capture both surface-level linguistic overlap and deeper semantic similarity. This ensures that evaluation does not merely reflect stylistic resemblance but instead rewards accurate semantics of the underlying time series phenomena. Refer to Appendix F for more details.

**numeric Fidelity Metrics.** Since TSC involves reporting exact or approximate numeric values, we introduce two tailored metrics to quantify numeric accuracy, both bounded within $[0, 1]$. The choice of the $5\%$ tolerance is discussed in Appendix F.2.

1. **Statistical Inference Accuracy.** While models are explicitly prompted to discuss descriptive statistics, they demonstrate varying abilities to accurately infer and verbalize statistics such as the mean, standard deviation, minimum, and maximum based on the raw time series and metadata. To evaluate this behavior, we report the percentage of captions in which these statistics are mentioned and fall within a $5\%$ relative error, using offline-computed true values. Importantly, captions are not penalized for omitting statistics; only wrongly reported values are considered errors. This metric primarily measures hallucination, favoring omission over incorrect numeric claims.

2. **Numeric Score.** For each ground truth caption, we extract all numeric values (excluding time-related ones like year or month) and search for the closest numeric value in the generated caption. A match is recorded if the closest value is within a $5\%$ relative tolerance. We compute *Accuracy* (mean of $1 - \min\{\text{relative\_error}, \text{tolerance}\}$) over all matched numbers), *Recall* (fraction of ground truth numbers matched), and a *Final Score* as a weighted combination: $\lambda_A \cdot \text{Accuracy} + \lambda_R \cdot \text{Recall}$, with $\lambda_A = 0.3$ and $\lambda_R = 0.7$ to emphasize recall over precision, as omitting critical numbers is more severe than minor numeric rounding imprecisions. While the previous metric targets numeric hallucinations, this one focuses on penalizing captions that omit numeric details.

## 4 EXPERIMENTS

We evaluate a broad range of VLMs on CaTS-Bench, covering both proprietary and open-source models, with the latter also tested after finetuning on our captioning training set (details in Appendix D). For TSC, we additionally consider a *program-aided* (PAL) model (Gao et al., 2023). All models are prompted with the same template-based format to ensure fair comparison, avoiding task- or architecture-specific prompt engineering. Appendix E provides the full model list and a description of PAL, while Appendix O outlines the human baseline that participated in our Q&A evaluation.

### 4.1 TIME SERIES CAPTIONING

To ensure fair comparison across the domains, we report macro-averaged scores for each metric, mitigating sample size imbalances, as some domains contain more data, and preventing any domain from disproportionately influencing the results. We benchmark leading VLMs on TSC using the

semi-synthetic and human-revisited captions separately as ground truth. Selected results are shown in Tables 3 and 4, with complete results in the Appendix G.

Table 3: Selected evaluation results of generated captions against human-revisited (HR) and semi-synthetic (SS) ground truths. **Bolded** and underlined scores denote first and second places.

| Category | Model | DeBERTa F1 | | SimCSE | | BLEU | | ROUGE-L | | METEOR | | Numeric | |
|---|---|---|---|---|---|---|---|---|---|---|---|---|---|
| | | HR | SS | HR | SS | HR | SS | HR | SS | HR | SS | HR | SS |
| Proprietary | Gemini 2.0 Flash | 0.665 | 0.688 | 0.856 | 0.858 | 0.079 | 0.137 | 0.248 | 0.318 | 0.221 | 0.279 | 0.634 | 0.677 |
| | Gemini 2.5 Pro | 0.657 | 0.668 | 0.857 | 0.845 | 0.069 | 0.088 | 0.236 | 0.267 | 0.247 | 0.284 | 0.681 | 0.714 |
| | Claude 3 Haiku | 0.658 | 0.682 | 0.853 | 0.856 | 0.064 | 0.112 | 0.241 | 0.291 | 0.236 | 0.300 | 0.601 | 0.623 |
| | GPT-4o | 0.661 | 0.681 | 0.863 | 0.865 | 0.071 | 0.112 | 0.233 | 0.284 | 0.236 | 0.296 | 0.627 | 0.644 |
| Pretrained | InternVL 2.5 38b | 0.664 | 0.688 | 0.871 | 0.868 | 0.072 | 0.129 | 0.244 | 0.305 | 0.255 | 0.331 | 0.659 | 0.685 |
| | LLaVA v1.6 | 0.627 | 0.650 | 0.824 | 0.820 | 0.052 | 0.086 | 0.215 | 0.259 | 0.233 | 0.287 | 0.455 | 0.517 |
| | LLaVA v1.6 34b | 0.639 | 0.655 | 0.821 | 0.825 | 0.060 | 0.094 | 0.221 | 0.265 | 0.232 | 0.285 | 0.547 | 0.560 |
| | Idefics 2 | 0.602 | 0.604 | 0.784 | 0.698 | 0.024 | 0.040 | 0.192 | 0.226 | 0.140 | 0.162 | 0.424 | 0.455 |
| | SmolVLM | 0.592 | 0.594 | 0.755 | 0.693 | 0.027 | 0.044 | 0.194 | 0.224 | 0.154 | 0.178 | 0.431 | 0.474 |
| | QwenVL | 0.619 | 0.643 | 0.821 | 0.890 | 0.049 | 0.082 | 0.209 | 0.249 | 0.214 | 0.261 | 0.445 | 0.504 |
| | QwenVL PAL | 0.664 | 0.685 | 0.864 | 0.843 | 0.066 | 0.108 | 0.237 | 0.292 | 0.226 | 0.282 | 0.564 | 0.613 |
| | Llama 3.2 V | 0.653 | 0.671 | 0.852 | 0.850 | 0.072 | 0.118 | 0.239 | 0.290 | 0.252 | 0.315 | 0.650 | 0.685 |
| | Gemma 3 27b | 0.648 | 0.667 | 0.863 | 0.863 | 0.065 | 0.085 | 0.222 | 0.263 | 0.257 | 0.309 | 0.641 | 0.668 |
| Finetuned | LLaVA v1.6 | **0.712** | 0.758 | **0.896** | 0.907 | **0.134** | 0.285 | **0.312** | 0.445 | **0.300** | **0.441** | **0.693** | 0.732 |
| | Idefics 2 | 0.711 | **0.759** | 0.894 | **0.908** | 0.132 | **0.290** | 0.309 | **0.452** | 0.298 | 0.437 | 0.691 | **0.733** |
| | InternVL-2.5 8b | 0.638 | 0.655 | 0.817 | 0.809 | 0.053 | 0.088 | 0.215 | 0.259 | 0.229 | 0.282 | 0.582 | 0.594 |
| | QwenVL | 0.703 | 0.643 | 0.892 | 0.790 | 0.126 | 0.082 | 0.302 | 0.249 | 0.297 | 0.260 | 0.683 | 0.504 |
| | SmolVLM | 0.604 | 0.613 | 0.817 | 0.781 | 0.051 | 0.091 | 0.228 | 0.269 | 0.220 | 0.265 | 0.556 | 0.643 |

**Semi-synthetic (SS) Captions as Ground Truth.** Our experiments show that finetuning substantially improves performance across most metrics. Proprietary models such as `GPT-4o` and *Gemini* generally outperform `Claude`. Among open-source models, finetuned `Idefics 2` and `LLaVA v1.6 Mistral` achieve strong gains, in some cases surpassing proprietary baselines, underscoring the effectiveness of finetuning for both linguistic quality and numeric precision. `QwenVL PAL` shows marked improvements over standard `QwenVL` and even takes the lead on statistical inference metrics (as shown in Table 4), highlighting code execution as a practical enhancement for tasks where numbers matter.

Given the semi-synthetic nature of ground truths in this experiment, we assessed the robustness of evaluation along two axes. First, to account for the stochasticity of LLM outputs, we repeated inference three times on $\sim 600$ test samples across five representative models; variance was vanishingly small (often $10^{-6}$; Appendix H.5), confirming that our single-run results are stable and reliable. Second, to test sensitivity to linguistic style, we paraphrased a subset of ground truth captions using multiple architecturally distinct LLMs while strictly preserving all factual content and numeric details, generating variants of ground truths differing only by linguistic style. The paraphrasing prompt is provided in Appendix N.3. Re-evaluating baseline outputs against these paraphrased captions as ground truth yielded model performance rankings largely consistent with those based on the original Gemini captions, with a mean Spearman Correlation of 0.9266

Table 4: Representative statistical inference scores under ground truths. E.g., *Mean* indicates statistical inference of the series mean. **Bolded** and underlined scores denote first and second places.

| Category | Model | Mean | | Max | | Min | |
|---|---|---|---|---|---|---|---|
| | | HR | SS | HR | SS | HR | SS |
| Proprietary | Gemini 2.0 Flash | 0.536 | 0.651 | 0.982 | 0.985 | 0.936 | 0.917 |
| | Gemini 2.5 Pro Prev. | 0.323 | 0.494 | 0.987 | **0.994** | **0.977** | **0.971** |
| | Claude 3 Haiku | 0.833 | 0.693 | 0.980 | 0.977 | 0.934 | 0.898 |
| | GPT-4o | 0.817 | 0.700 | **0.992** | 0.990 | 0.938 | 0.921 |
| Pretrained | InternVL 2.5 38b | 0.858 | 0.784 | 0.982 | 0.966 | 0.930 | 0.887 |
| | LLaVA v1.6 Mistral | 0.667 | 0.644 | 0.871 | 0.864 | 0.751 | 0.743 |
| | LLaVA v1.6 34b | 0.410 | 0.445 | 0.817 | 0.843 | 0.727 | 0.698 |
| | Idefics 2 | 0.806 | 0.616 | 0.891 | 0.903 | 0.840 | 0.806 |
| | QwenVL | 0.656 | 0.565 | 0.795 | 0.822 | 0.678 | 0.657 |
| | QwenVL PAL | **0.973** | **0.903** | 0.985 | 0.980 | 0.978 | 0.942 |
| | Llama 3.2 Vision | 0.467 | 0.594 | 0.956 | 0.952 | 0.895 | 0.877 |
| | Gemma 3 27b | 0.734 | 0.694 | 0.978 | 0.968 | 0.904 | 0.864 |
| Finetuned | LLaVA v1.6 Mistral | 0.928 | 0.828 | 0.987 | 0.976 | **0.981** | 0.926 |
| | Idefics 2 | 0.958 | 0.885 | 0.988 | 0.985 | 0.967 | **0.927** |
| | InternVL 2.5 (8b) | 0.750 | 0.597 | 0.830 | 0.904 | 0.734 | 0.779 |
| | QwenVL | 0.952 | 0.565 | 0.973 | 0.822 | 0.963 | 0.657 |
| | SmolVLM | 0.640 | 0.590 | 0.914 | 0.898 | 0.772 | 0.777 |

and metric-specific correlations shown in Table 11 (full discussion in H.3). These results corroborate that our evaluation framework is stable and reliably gauges caption quality rather than biased surface-level stylistic alignment.

**Human-revisited (HR) Captions as Ground Truth.** We repeat the evaluation using human-revisited captions as ground truth, further confirming the benefits of finetuning. Open-source models like `Idefics 2` and `LLaVA v1.6 Mistral` gain substantially in text quality and numeric accuracy, often surpassing proprietary baselines on linguistic metrics and nearing them on numeric ones. Proprietary models such as `GPT-4o` and *Gemini* still lead on some language-focused metrics, but their advantage shrinks when finetuned open-source models are included. Meanwhile, the *PAL* model excels in statistical inference thanks to code execution. Overall, these results confirm that finetuning not only enhances average performance but also improves numeric reliability, positioning open-source models as strong contenders when paired with targeted adaptation.

### 4.2 Q&A Tasks

Figure 3 summarizes model performance on our Q&A tasks, while Table 17 provides detailed results. Performance is highly variable, and even proprietary models occasionally fail to exceed random chance on some tasks.

No model consistently dominates across all categories. Models handle binary-choice time series comparisons better, likely due to the narrower range of options. Matching a time series to a caption is harder than the reverse, and plot matching is the most challenging, highlighting a key VLM weakness: linking numeric patterns with visual features. Proprietary models (`GPT-4o`, `Gemini 2.0 Flash`) lead, while among open-source models, `Phi-4 M.I.` excels in time series and statistical reasoning. Finetuning on TSC yields mixed results: some models (e.g., `Phi-4 M.I.`, `Idefics 2`) gain in specific sub-tasks, while others drop in performance. Notably, finetuning often fails to improve Q&A accuracy, likely due to task misalignment and catastrophic forgetting. As Table 17 shows, humans achieve the highest overall scores, though top models sometimes outperform them on distraction-prone tasks. Notably, all models perform near-random on plot matching, whereas humans score nearly perfectly. Despite the tasks' apparent simplicity, they reveal fundamental limitations in VLMs' temporal reasoning capabilities which suggests the need to address basic time series understanding before tackling more complex applications.

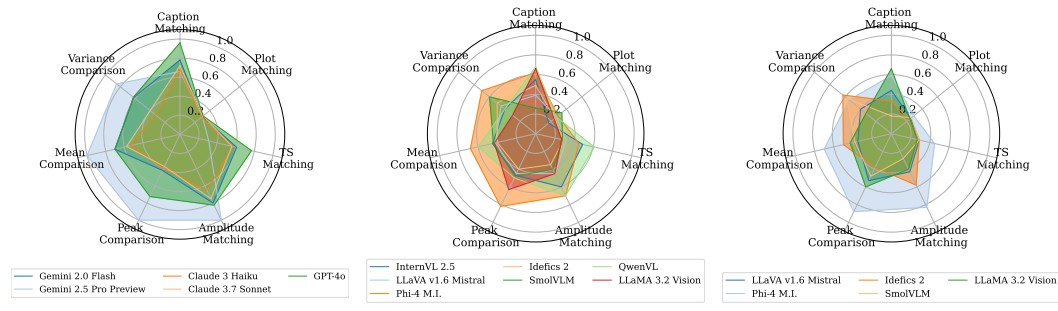

|        (a) Proprietary VLMs        |        (b) Pretrained VLMs        |        (c) Finetuned VLMs        |

Figure 3: Model accuracy across Q&A sub-tasks. Proprietary models perform best, pretrained models lag behind, and finetuned models struggle across all tasks.

### 4.3 Role of the Visual Modality

**Visual Modality Ablation.** We perform a modality removal experiment by stripping away the time series plot and providing only the associated textual metadata and the numeric values of the time series. This quantifies the contribution of the visual channel and enables a better understanding of the model's captioning performance. We evaluate a selected subset of pretrained baselines to assess their intrinsic reliance on vision. Full results can be found in Appendix I.1.

Our experiments suggest that the additional contribution of the visual modality to caption quality is insignificant for most models. As shown in Figure 4, most models show only marginal performance drops, or even slight gains, when the time series plot is removed, suggesting a strong dependence on

textual priors over visual understanding. In particular, models such as `Idefics2`, `Phi-4 M.I.`, and `InternVL` perform better in text-only settings on most metrics, hinting that generation is largely driven by language pretraining or instruction tuning rather than true visual interpretation.

Figure 4: Performance deltas between VL (vision-language input) and L (text-only input). Each cell shows $\Delta = \text{VL} - \text{L}$. Blue indicates better performance due to the visual input; red the opposite.

Models such as `QwenVL`, `LLaVA 1.6` and `Claude 3 Haiku` maintain strong performance with visual input, but the performance gap ($\Delta$) remains modest, underscoring the underuse of plot-based information. Interestingly, the numeric score tends to decline when visual input is removed, hinting at weak but present reliance on the plot for numeric reasoning. These results point to a subtle yet important misalignment: models are exposed to visual data but often fail to meaningfully reason with it. This phenomenon is not limited to line plots, as discussed in I.3, even more expressive visual forms (e.g., Gramian Angular Fields and recurrence plots) fail to trigger visual reasoning of current VLMs in TSC.

**Visual Attention Analysis.** To better understand how VLMs process plots during caption generation, we examined their attention maps. The analysis revealed minimal visual grounding: models concentrated predominantly on textual elements in the plots (e.g., axis labels and titles), with limited evidence of attending to the actual line trends. Attention to visual patterns was sporadic, weak, and inconsistent, suggesting that learned parameters largely disregard visual cues in favor of textual priors. This qualitative evidence highlights the gap between nominal multimodal input and actual integration. Full results are reported in Appendix I.2 and Figure 7.

It is important to note that the under-utilization of visual inputs observed in our experiments is not a limitation of CaTS-Bench itself, but rather a reflection of current VLM capabilities. The benchmark explicitly provides both time series plots and rich metadata, creating ample opportunity for multimodal reasoning. That most models default to textual priors instead of leveraging visual signals highlights a critical gap in the field. We view this as an opportunity for future research: developing models that better integrate plot-based information with textual and numeric cues to advance the broader goal of genuine multimodal understanding in time series analysis.

## 5 CONCLUSION

We introduced CaTS-Bench, the first large-scale, multimodal benchmark for context-aware time series captioning and reasoning. Built from 11 diverse real-world datasets, it combines numeric series, metadata, visual plots, and validated captions to provide a challenging testbed beyond synthetic or narrow benchmarks. A key contribution is not only the benchmark itself, but also the scalable data curation pipeline we developed to generate high-quality captions. This pipeline leverages an oracle LLM anchored in metadata, rigorous verification through factual checks, diversity analyses, and a complementary human-revisited subset, making it both scalable and extensible to new domains. Our evaluation of leading VLMs revealed both progress and limitations. Finetuning greatly improves open-source models, enhancing fluency and numeric fidelity, while proprietary models show stronger performance overall. A consistent weakness lies in multimodal grounding: models largely ignore visual inputs, with plot matching emerging as the most difficult task. These findings reveal a critical gap in multimodal alignment and point toward the urgent need for models that can genuinely integrate numeric, textual, and visual cues. By releasing CaTS-Bench together with its evaluation suite, we provide the community with not only a rigorous foundation for advancing time series reasoning, but also a practical methodology for generating reliable, context-rich captions at scale, paving the way for more robust multimodal understanding in the future.

ETHICAL STATEMENT

The development of CaTS-Bench was guided by a commitment to ethical research practices. All datasets used in this work are publicly available and do not contain personally identifiable information (PII). The domains, such as climate, public health, and agriculture, were chosen for their public relevance and data accessibility. Our use of an oracle LLM to generate semi-synthetic reference captions was a deliberate design choice to ensure scalability, particularly for a subjective task like captioning, where a single ground-truth is ill-defined. We have taken extensive measures to validate the quality, factual accuracy, and diversity of these semi-synthetic captions, as detailed in Section 3.2 and Appendix H, to mitigate the risk of propagating systemic biases from the oracle model. Our human-revisited test set is also an attempt to further ensure evaluation reliability. For our human evaluation studies, all participation was voluntary. We obtained informed consent from all participants, who were university students. The study's purpose was clearly communicated, and all responses were collected anonymously to protect participant privacy, as shown in an example of a consent form in Appendix O.

REPRODUCIBILITY STATEMENT

To ensure the reproducibility of our research, all components of our work will be made publicly available upon publication.

1. **Data:** The complete CaTS-Bench dataset, including the numeric time series, metadata, generated plots, oracle-generated and human-revisited captions, and the diagnostic Q&A suite, are released at `https://huggingface.co/datasets/a9f3c7e2/CaTSBench`.

2. **Code:** We will release the source code for the entire data curation pipeline, model finetuning scripts, and the evaluation suite. The code will be hosted in a public repository to allow for complete replication of our results and to facilitate future research.

3. **Models and Environment:** All open-source models used in our experiments are explicitly named with version details provided in Appendix E. For proprietary models, we specify the exact model endpoints used at the time of the experiments. Detailed finetuning hyperparameters and hardware specifications are documented in Appendix D.

4. **Evaluation:** Our evaluation protocol relies on standard, well-established linguistic metrics and novel metrics that are precisely defined in F. All prompts used for caption generation, quality verification, and LLM-based scoring are provided in Appendix N to ensure that our evaluation can be replicated consistently. Furthermore, we conducted a robustness check (Appendix H.5), which demonstrated minimal variance across multiple runs, confirming the stability of our results.

LLM USAGE STATEMENT

Large Language Models played a central role in multiple stages of this work.

1. LLMs were employed as **data generators**, producing semi-synthetic captions that serve as ground truth references in CaTS-Bench.

2. LLMs were employed as **data extractors**, for example to parse statistical claims from captions during our evaluation analyses.

3. LLMs, more precisely VLMs, served as **baselines** in our experiments as captioning models for evaluation.

4. LLMs were employed as a **writing assist tool** to polish the presentation of the paper, while the authors retain full responsibility for all content.

Importantly, LLMs did not contribute to research ideation or decision-making. All factual claims, analyses, and conclusions are the responsibility of the authors.

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

# Appendix

## Table of Contents

## A   LIMITATIONS AND FUTURE WORK

While CaTS-Bench represents a significant step toward multimodal, context-aware time series understanding, it also has limitations that suggest clear avenues for future work. First, the majority of captions are semi-synthetic and generated by a single oracle model (`Gemini 2.0 Flash`). Our validation studies confirm their factual reliability and linguistic diversity, with no clear evidence of bias; however, reliance on a single oracle may still introduce subtle, hidden modeling biases. Although we release a subset of data paraphrased by different LLMs, its limited scale points to future iterations of CaTS-Bench where semi-synthetic captions are fully generated from a broader pool of LLMs, thereby reducing such biases and better reflecting diversity in expression. Second, although we provide a curated human-revisited subset of captions, the scale of fully human-authored or expert-verified content remains limited. We acknowledge that captions written by human experts are often more insightful, and future work should incorporate such expert-written captions to enhance depth and interpretability. At the same time, human authors introduce their own stylistic biases, which should be considered when designing and evaluating the benchmark. Expanding this component, potentially by involving multiple domain experts in economics, healthcare, or climate data, would further strengthen the benchmark's robustness and credibility.

Overall, we view CaTS-Bench as a scalable foundation rather than a fixed resource, with ample room to grow through multi-oracle captioning, richer human input, and extended coverage of temporal reasoning tasks.

## B   SOURCE DATASETS

1. **Air Quality** – Hourly air pollution data from 453 Indian cities (2010–2023), covering 30+ parameters including $PM_{2.5}$, $NO_x$, CO, and $SO_2$, compiled from CPCB Jha (2023).

2. **Border Crossing** – Monthly inbound border crossing counts at U.S.-Mexico and U.S.-Canada ports, disaggregated by transport mode and collected by U.S. Customs and Border Protection U.S. Department of Transportation (n.d.).

3. **Crime** – Incident-level crime reports in Los Angeles from 2020 onward, provided by LAPD OpenData and updated biweekly, including NIBRS-compliant records of Los Angeles (n.d.).

4. **Demography** – Annual global indicators from the UN and World Bank (2000–2021) covering population growth, fertility, life expectancy, death rates, and median age to assess patterns of demographic change and collapse Aziz (1985).

5. **Injury** – Annual counts of fatal and severe road traffic injuries in California (2002–2010), disaggregated by transport mode and geography, from CDPH's Healthy Communities Indicators of Public Health (n.d.).

6. **COVID** – Global daily COVID-19 case and death counts (2020), compiled by ECDC, covering over 200 countries with population-adjusted metrics European Centre for Disease Prevention and Control (2024).

7. **CO₂** – National-level per capita $CO_2$ emissions and GDP trends from Our World in Data, adjusted for trade (consumption-based), spanning 1990–2023 Ritchie (2021).

8. **Calories (Diet)** – Food supply and caloric intake patterns from FAO Food Balance Sheets Food and Agriculture Organization of the United Nations (2024).

9. **Walmart** – Weekly sales data from 45 Walmart stores (2010–2012), enriched with features like temperature, fuel price, CPI, unemployment rate, and holiday flags Hassan (2020).

10. **Retail** – Transactional records from a UK-based online gift retailer (2010–2011), capturing item-level purchases, cancellations, and customer behavior Chen (2015).

11. **Agriculture** – Annual agricultural total factor productivity (TFP) indices from USDA for 1961–2022, covering outputs and inputs like land, labor, capital, and materials across countries USDA Economic Research Service (2024).

## C   TIME SERIES SEGMENT CROPPING

Our cropping strategy balances diversity with consistency across datasets. Many source time series (e.g., 50 years of hourly CO2 emissions) are too long to process directly, so we sample random windows of variable lengths. Each window length is drawn from a dataset-specific range $[min, max]$, with the maximum based on the original series length. This ensures that cropped windows preserve the scale and structure of the data while introducing sufficient variability for training and evaluation. We summarize these rules in Table 5 below.

Table 5: Minimum and maximum segment lengths for each dataset.

| Source Dataset | Min Length | Max Length |
| --- | --- | --- |
| Air Quality, Crime, Border Crossing, $CO_2$, Walmart, Agriculture | 5 | $\min(150,\ 5 + \lfloor \frac{\text{original\_length}}{8} \rfloor)$ |
| Demography, Online Retail | 5 | original_length |
| Road Injuries | 3 | $\min(3,\ 0.2 \times \text{original\_length})$ |
| COVID | 5 | $\min(150,\ 5 + \lfloor \frac{\text{original\_length}}{5} \rfloor)$ |
| Calories | 5 | $\min(6,\ 0.2 \times \text{original\_length})$ |

## D   HARDWARE AND SETTINGS

All experiments were conducted on a high-performance computing node featuring two *AMD EPYC 7453* processors, providing a total of 56 logical CPUs, and 125 GB of RAM (with over 117 GB available during runtime). For GPU acceleration, the system includes eight *NVIDIA A100* GPUs - six PCIe 80 GB models and two PCIe 40 GB models - along with an ASPEED graphics controller used for display purposes. This configuration offers ample computational and memory resources suitable for mid- to large-scale deep learning training and inference. The models we finetune range in size from 2 billion to 11 billion parameters, with finetuning times spanning from a few hours to a day.

For finetuning, we adopt a unified training strategy guided by best practices in instruction tuning for multimodal inputs. All models are trained using the AdamW optimizer with a cosine learning rate scheduler and a base learning rate of $2 \times 10^{-5}$. We apply gradient accumulation to simulate a larger batch size. Mixed precision training and gradient checkpointing are enabled for memory efficiency. Low Rank Adaptation (LoRA) is used to adapt large models by tuning a small subset of parameters, while keeping the rest of the model frozen or partially frozen. To ensure deterministic and focused generation, we use a temperature of $0.3$ during inference across all evaluated models. Each model is finetuned using a structured JSONL dataset comprising time series plot images and corresponding image-grounded chat-style conversations. We preprocess data with each model's native processor and apply minimal resizing to maintain fidelity in the visual input. Special care is taken to exclude padding and <image> tokens from loss computation by assigning them an ignore index.

Table 6: Configurations

| Hyperparameter | Value |
| --- | --- |
| Batch size | 4 |
| Grad. Acc. | 12 |
| Epochs | 3 |
| Learning Rate | $2 \times 10^{-5}$ |
| Scheduler | Cosine |
| Optimizer | AdamW |
| Precision | `bf16` |
| LoRA rank | 8 or 16 |
| Dropout | 0.05 |
| Image res. | 224–560 |

## E   BASELINE MODELS

We evaluate `Gemini 2.0 Flash` and `Gemini 2.5 Pro Preview` (Team et al., 2023), `Claude 3 Haiku` and `Claude 3.7 Sonnet` (Anthropic, 2024), `GPT-4o` (Achiam et al., 2023), `InternVL 2.5` (8b & 38b) (Chen et al., 2024b), `LLaVA v1.6 Mistral 7b` (default) and `34b` (Liu et al., 2023), `Phi-4 Multimodal Instruct 5.6b` (Abdin et al., 2024), `Idefics 2 (8b)` (Laurençon et al., 2024), `SmolVLM (2b)` (Marafioti et al., 2025), `QwenVL`

(7b) (Bai et al., 2023), `Llama 3.2 Vision (11b)` (Grattafiori et al., 2024), and `Gemma 3 (12b & 27b)` (Team et al., 2025) for both TSC and Q&A tasks.

TSC requires precise numeric reasoning alongside text generation, making it suitable for program-aided language (PAL) models (Gao et al., 2023). Hence, we also evaluate `QwenVL 32b` by prompting it to generate a Python program that outputs the full time series caption. The program is executed in Python, and its return value is taken as the caption. Most ( 90%) generated programs succeed on the first attempt; if a program fails, we increase the token limit and regenerate until successful. The full prompt example can be found in Appendix N.4.

# F  EVALUATION METRICS

## F.1  LINGUISTIC METRICS

**DeBERTa Score**  The DEBERTA SCORE is a contextual similarity metric based on cosine similarity between contextual embeddings of tokens in the candidate ($c$) and reference ($r$) captions. Given token embeddings from the DeBERTa encoder, the metric computes token-level precision, recall, and F1:

$$\text{F1}_{\text{DeBERTa}} = \frac{2 \cdot P \cdot R}{P + R}, \quad P = \frac{1}{|c|} \sum_{i \in c} \max_{j \in r} \cos(\mathbf{e}_i, \mathbf{e}_j), \quad R = \frac{1}{|r|} \sum_{j \in r} \max_{i \in c} \cos(\mathbf{e}_j, \mathbf{e}_i) \quad (1)$$

where $\mathbf{e}_i$ and $\mathbf{e}_j$ are the contextual embeddings of candidate and reference tokens, respectively.

**BLEU**  BLEU evaluates n-gram overlap between a candidate caption and reference using precision with a brevity penalty to discourage short outputs:

$$\text{BLEU} = \text{BP} \cdot \exp\left(\sum_{n=1}^{N} w_n \log p_n\right), \quad \text{BP} = \begin{cases} 1, & \text{if } c > r \\ e^{1-r/c}, & \text{if } c \leq r \end{cases} \quad (2)$$

where $p_n$ is the modified precision for $n$-grams, $w_n$ are weights (usually uniform), $c$ is candidate length, and $r$ is reference length.

**ROUGE-L**  ROUGE-L measures fluency via the length of the longest common subsequence (LCS) between candidate and reference:

$$\text{ROUGE-L}_{\text{F1}} = \frac{(1 + \beta^2) \cdot \text{LCS}}{r + c}, \quad \text{LCS} = \text{LongestCommonSubsequence}(r, c) \quad (3)$$

where $\beta$ balances recall and precision (often $\beta = 1$), and $r$ and $c$ are the reference and candidate lengths.

**METEOR**  METEOR aligns unigrams using exact matches, stems, synonyms, and paraphrases. It then computes an F-score and applies a fragmentation penalty:

$$\text{METEOR} = F_{\text{mean}} \cdot (1 - \text{Pen}), \quad F_{\text{mean}} = \frac{10 \cdot P \cdot R}{R + 9P}, \quad \text{Pen} = 0.5 \left(\frac{\text{chunks}}{\text{matches}}\right)^3 \quad (4)$$

where $P$ and $R$ are unigram precision and recall, and chunks refers to non-contiguous matched segments.

**SimCSE**  SIMCSE computes semantic similarity at the sentence level using cosine similarity between sentence embeddings:

$$\text{SimCSE}(c, r) = \cos(\mathbf{h}_c, \mathbf{h}_r) = \frac{\mathbf{h}_c \cdot \mathbf{h}_r}{\|\mathbf{h}_c\|\|\mathbf{h}_r\|} \tag{5}$$

where $\mathbf{h}_c$ and $\mathbf{h}_r$ are candidate and reference sentence embeddings, produced by a contrastively trained RoBERTa encoder.

## F.2  TOLERANCE DESIGN IN NUMERIC METRICS

We adopt a $5\%$ relative tolerance for both numeric metrics, as it is a widely accepted threshold in numeric evaluation across data science and time series literature. This value balances sensitivity and robustness: it is tight enough to catch meaningful deviations from the true value, ensuring that significant errors are penalized, yet lenient enough to accommodate minor variations due to rounding, numeric precision, or natural approximations in model-generated captions. By using this standard threshold, our evaluation aligns with common practice while focusing on practically relevant numeric accuracy.

## G  FULL TSC RESULTS

Here, we report the full evaluation results for time series captioning, using human-revisited and semi-synthetic captions as ground truth. See Tables 7 and 8 respectively.

Table 7: Evaluation of generated captions against the human-revisited ground truths. Numeric: numeric score. Mean/STD/Max/Min refer to statistical inference accuracy. **Bolded** and underlined scores denote first and second places.

| Model | DeBERTa F1 | SimCSE | BLEU | ROUGE-L | METEOR | Mean | STD | Max | Min | Numeric |
|---|---|---|---|---|---|---|---|---|---|---|
| **Proprietary** | | | | | | | | | | |
| Gemini 2.0 Flash | 0.6645 | 0.8558 | 0.0793 | 0.2475 | 0.2205 | 0.5357 | - | 0.9823 | 0.9363 | 0.6335 |
| Gemini 2.5 Pro Prev. | 0.6568 | 0.8570 | 0.0690 | 0.2363 | 0.2468 | 0.3229 | - | 0.9871 | 0.9771 | 0.6805 |
| Claude 3 Haiku | 0.6580 | 0.8525 | 0.0640 | 0.2405 | 0.2358 | 0.8333 | - | 0.9797 | 0.9339 | 0.6007 |
| GPT-4o | 0.6605 | 0.8632 | 0.0705 | 0.2328 | 0.2355 | 0.8167 | - | **0.9921** | 0.9379 | 0.6268 |
| **Pretrained** | | | | | | | | | | |
| InternVL 2.5 (8b) | 0.6418 | 0.8411 | 0.0510 | 0.2093 | 0.2153 | 0.7813 | 0.0000 | 0.9495 | 0.8186 | 0.5812 |
| InternVL 2.5 (38b) | 0.6640 | 0.8707 | 0.0723 | 0.2442 | 0.2550 | 0.8581 | 0.0000 | 0.9820 | 0.9297 | 0.6585 |
| LLaVA v1.6 Mistral | 0.6268 | 0.8243 | 0.0525 | 0.2145 | 0.2333 | 0.6667 | 0.0000 | 0.8714 | 0.7505 | 0.4548 |
| LLaVA v1.6 34b | 0.6388 | 0.8205 | 0.0595 | 0.2210 | 0.2317 | 0.4103 | 0.5588 | 0.8170 | 0.7274 | 0.5465 |
| Phi-4 M.I. | 0.6168 | 0.8196 | 0.0450 | 0.2355 | 0.1945 | 0.4554 | 0.2500 | 0.9227 | 0.8926 | 0.5533 |
| Idefics 2 | 0.6023 | 0.7838 | 0.0235 | 0.1918 | 0.1400 | 0.8056 | 0.3854 | 0.8908 | 0.8401 | 0.4238 |
| SmolVLM | 0.5918 | 0.7552 | 0.0273 | 0.1935 | 0.1538 | 0.9050 | 0.5278 | 0.8972 | 0.7606 | 0.4308 |
| QwenVL | 0.6185 | 0.8205 | 0.0490 | 0.2088 | 0.2140 | 0.6563 | - | 0.7947 | 0.6776 | 0.4450 |
| QwenVL PAL | 0.6643 | 0.8643 | 0.0660 | 0.2373 | 0.2260 | **0.9730** | 0.8571 | 0.9848 | **0.9784** | 0.5638 |
| Llama 3.2 Vision | 0.6527 | 0.8520 | 0.0715 | 0.2390 | 0.2515 | 0.4667 | - | 0.9562 | 0.8952 | 0.6503 |
| Gemma 3 12b | 0.6568 | 0.8692 | 0.0713 | 0.2350 | 0.2633 | 0.8148 | **1.0000** | 0.9576 | 0.8956 | 0.6598 |
| Gemma 3 27b | 0.6480 | 0.8634 | 0.0650 | 0.2220 | 0.2565 | 0.7341 | - | 0.9780 | 0.9035 | 0.6407 |
| **Finetuned** | | | | | | | | | | |
| InternVL 2.5 (8b) | 0.6378 | 0.8171 | 0.0533 | 0.2148 | 0.2285 | 0.7500 | 0.0000 | 0.8302 | 0.7339 | 0.5815 |
| LLaVA v1.6 Mistral | **0.7123** | **0.8964** | **0.1335** | **0.3123** | **0.3003** | 0.9284 | 0.1746 | 0.9866 | 0.9806 | **0.6932** |
| Phi-4 M.I. | 0.6485 | 0.8501 | 0.0638 | 0.2393 | 0.2295 | 0.6250 | 0.2500 | 0.9655 | 0.9184 | 0.5860 |
| Idefics 2 | 0.7108 | 0.8942 | 0.1323 | 0.3085 | 0.2975 | 0.9580 | 0.3453 | 0.9875 | 0.9665 | 0.6912 |
| SmolVLM | 0.6035 | 0.8168 | 0.0513 | 0.2275 | 0.2198 | 0.6399 | 0.1250 | 0.9139 | 0.7721 | 0.5558 |
| QwenVL | 0.7032 | 0.8920 | 0.1263 | 0.3020 | 0.2965 | 0.9520 | 0.2430 | 0.9731 | 0.9634 | 0.6825 |
| Llama 3.2 Vision | 0.6470 | 0.8518 | 0.0693 | 0.2328 | 0.2483 | 0.5876 | 0.5000 | 0.9585 | 0.8961 | 0.6273 |

Table 8: Evaluation of generated captions against semi-synthetic ground truths. Numeric: numeric score. Mean/STD/Max/Min refer to statistical inference accuracy. **Bolded** and underlined scores denote first and second places.

| Model | DeBERTa F1 | SimCSE | BLEU | ROUGE-L | METEOR | Mean | STD | Max | Min | Numeric |
|---|---|---|---|---|---|---|---|---|---|---|
| **Proprietary** | | | | | | | | | | |
| Gemini 2.0 Flash | 0.688 | 0.858 | 0.137 | 0.318 | 0.279 | 0.651 | 0.916 | 0.985 | 0.917 | 0.677 |
| Gemini 2.5 Pro Prev. | 0.668 | 0.845 | 0.088 | 0.267 | 0.284 | 0.494 | 0.667 | **0.994** | **0.971** | 0.714 |
| Claude 3 Haiku | 0.682 | 0.856 | 0.112 | 0.291 | 0.300 | 0.693 | 0.735 | 0.977 | 0.898 | 0.623 |
| GPT-4o | 0.681 | 0.865 | 0.112 | 0.284 | 0.296 | 0.700 | 0.778 | 0.990 | 0.921 | 0.644 |
| **Pretrained** | | | | | | | | | | |
| InternVL 2.5 (8b) | 0.659 | 0.794 | 0.081 | 0.247 | 0.260 | 0.610 | **0.920** | 0.949 | 0.794 | 0.589 |
| InternVL 2.5 (38b) | 0.688 | 0.868 | 0.129 | 0.305 | 0.331 | 0.784 | 0.640 | 0.966 | 0.887 | 0.685 |
| LLaVA v1.6 Mistral | 0.650 | 0.820 | 0.086 | 0.259 | 0.287 | 0.644 | 0.611 | 0.864 | 0.743 | 0.517 |
| LLaVA v1.6 34b | 0.655 | 0.825 | 0.094 | 0.265 | 0.285 | 0.445 | 0.550 | 0.843 | 0.698 | 0.560 |
| Phi-4 M.I. | 0.624 | 0.797 | 0.074 | 0.274 | 0.239 | 0.457 | 0.443 | 0.942 | 0.859 | 0.583 |
| Idefics 2 | 0.604 | 0.698 | 0.040 | 0.226 | 0.162 | 0.616 | 0.368 | 0.903 | 0.806 | 0.455 |
| SmolVLM | 0.594 | 0.693 | 0.044 | 0.224 | 0.178 | 0.747 | 0.446 | 0.864 | 0.705 | 0.474 |
| QwenVL | 0.643 | 0.890 | 0.082 | 0.249 | 0.261 | 0.565 | 0.257 | 0.822 | 0.657 | 0.504 |
| QwenVL PAL | 0.685 | 0.843 | 0.108 | 0.292 | 0.282 | **0.903** | 0.549 | 0.980 | 0.942 | 0.613 |
| Llama 3.2 Vision | 0.671 | 0.850 | 0.118 | 0.290 | 0.315 | 0.594 | 0.666 | 0.952 | 0.877 | 0.685 |
| Gemma 3 12b | 0.676 | 0.867 | 0.097 | 0.279 | 0.317 | 0.653 | 0.578 | 0.957 | 0.879 | 0.673 |
| Gemma 3 27b | 0.667 | 0.863 | 0.085 | 0.263 | 0.309 | 0.694 | 0.900 | 0.968 | 0.864 | 0.668 |
| **Finetuned** | | | | | | | | | | |
| InternVL 2.5 (8b) | 0.655 | 0.809 | 0.088 | 0.259 | 0.282 | 0.597 | 0.464 | 0.904 | 0.779 | 0.594 |
| LLaVA v1.6 Mistral | 0.758 | 0.907 | 0.285 | 0.445 | **0.441** | 0.828 | 0.294 | 0.976 | 0.926 | 0.732 |
| Phi-4 M.I. | 0.662 | 0.821 | 0.010 | 0.285 | 0.279 | 0.645 | 0.641 | 0.965 | 0.877 | 0.607 |
| Idefics 2 | **0.759** | **0.908** | **0.290** | **0.452** | 0.437 | 0.885 | 0.379 | 0.985 | 0.927 | **0.733** |
| SmolVLM | 0.613 | 0.781 | 0.091 | 0.269 | 0.265 | 0.590 | 0.297 | 0.898 | 0.777 | 0.643 |
| QwenVL | 0.643 | 0.790 | 0.082 | 0.249 | 0.260 | 0.565 | 0.257 | 0.822 | 0.657 | 0.504 |
| Llama 3.2 Vision | 0.667 | 0.844 | 0.111 | 0.283 | 0.310 | 0.502 | 0.619 | 0.955 | 0.867 | 0.668 |

# H QUALITY VALIDATION

## H.1 MANUAL VERIFICATION OF SEMI-SYNTHETIC GROUND TRUTH CAPTIONS

To assess the reliability of semi-synthetic captions used as ground truth, we manually verified statistical and trend claims in around $2.9k$ captions of the test set under a three-tier scoring system: *exact* (within $\pm0.05$ of the true value), *near* (within $\pm0.1$), and *incorrect*.

Table 9: Manual verification of SS captions. Accuracy remains high across all categories.

| Task | Feature/Type | Occurences Verified | Accuracy |
|---|---|---|---|
| Statistical | mean / average | 570 | 0.980 |
| | minimum / dip | 310 | 0.994 |
| | standard deviation | 123 | 1.000 |
| | maximum / peak | 217 | 0.994 |
| Trend | Upward | 467 | 0.980 |
| | Downward | 328 | 0.997 |
| | Stability | 45 | 0.970 |
| | Fluctuation | 87 | 0.974 |
| Historical (Trends + Stats) | mean / average | 435 | 0.980 |
| | standard deviation | 26 | 1.000 |
| | maximum / peak | 95 | 0.994 |
| | minimum / dip | 12 | 1.000 |
| | norms | 164 | 0.980 |
| **Total / Avg.** | | **2879** | **0.986** |

**Method** Each claim was extracted and checked against the underlying time series metadata. We extracted statistical claims and trend patterns from captions using structured keyword clustering across selected statistical categories (mean/average, min/max, standard deviation) and trend keywords (increasing, decreasing, plateau, fluctuation). We (1) identified terms using keyword matching, (2) extracted ground truth values therein, and (3) created verification sheets comparing claims against actual metadata for manual verification. Then, we scored each claim by comparing statements against ground truth data and verifying that trend claims matched actual trajectories.

Accuracy across both statistical descriptors and trend types is consistently high, confirming that **oracle-generated captions provide factually reliable reference annotations**.

## H.2 Human Study on Detectability

We conducted a blind study with 35 participants, each reviewing 11 captions (half written by humans, half by Gemini, using the same context information). Participants labeled each as human or AI-written but achieved only $41.1\%$ accuracy, essentially random, suggesting **Gemini's captions were indistinguishable from human ones**. Participation form with guidelines is in Appendix O.

## H.3 Robustness of Evaluation - Paraphrasing Experiment

A legitimate concern when using a single LLM to generate reference captions is the potential for evaluation bias towards the specific linguistic style of that model. To ensure that CaTS-Bench evaluates generalizable time series understanding capabilities rather than facility in mimicking a Gemini's linguistic style, we conducted a robustness experiment.

Table 10: Evaluation of generated captions across paraphrased/original ground truths.

| Model | GT Style | Embedding | | N-gram | | | Numeric | Stat. Inference | | |
|---|---|---|---|---|---|---|---|---|---|---|
| | | SimCSE | DeBERTa | METEOR | ROUGE-L | BLEU | Numeric | Mean | Max | Min |
| Gemini 2.0 Flash | GPT-4o Phr. | 0.8707 | 0.6803 | 0.2313 | 0.2605 | 0.0820 | 0.6715 | 0.3333 | 0.9823 | 0.9377 |
| | Gemma Phr. | 0.8635 | 0.6748 | 0.2068 | 0.2443 | 0.0605 | 0.6578 | 0.5357 | 0.9823 | 0.9374 |
| | Llama Phr. | 0.8680 | 0.6715 | 0.2080 | 0.2445 | 0.0710 | 0.6745 | 0.3750 | 0.9807 | 0.9351 |
| | Original | 0.8716 | 0.6860 | 0.2720 | 0.3068 | 0.1315 | 0.6802 | 0.3750 | 0.9823 | 0.9377 |
| GPT-4o | GPT-4o Phr. | 0.8752 | 0.6740 | 0.2583 | 0.2488 | 0.0845 | 0.6773 | 0.8000 | 0.9921 | 0.9393 |
| | Gemma Phr. | 0.8645 | 0.6665 | 0.2250 | 0.2268 | 0.0578 | 0.6673 | 0.8167 | 0.9921 | 0.9393 |
| | Llama Phr. | 0.8726 | 0.6678 | 0.2295 | 0.2380 | 0.0725 | 0.6673 | 0.8000 | 0.9921 | 0.9379 |
| | Original | 0.8773 | 0.6785 | 0.2880 | 0.2785 | 0.1048 | 0.6558 | 0.8000 | 0.9921 | 0.9379 |
| Claude 3 Haiku | GPT-4o Phr. | 0.8636 | 0.6720 | 0.2497 | 0.2480 | 0.0693 | 0.6383 | 0.7500 | 0.9797 | 0.9338 |
| | Gemma Phr. | 0.8563 | 0.6665 | 0.2263 | 0.2295 | 0.0495 | 0.6223 | 0.8000 | 0.9781 | 0.9321 |
| | Llama Phr. | 0.8598 | 0.6678 | 0.2320 | 0.2473 | 0.0605 | 0.6285 | 0.8333 | 0.9797 | 0.9339 |
| | Original | 0.8683 | 0.6795 | 0.2873 | 0.2870 | 0.1038 | 0.6333 | 0.7500 | 0.9797 | 0.9338 |
| Idefics 2 | GPT-4o Phr. | 0.7952 | 0.6113 | 0.1463 | 0.2035 | 0.0243 | 0.3893 | 0.8056 | 0.8908 | 0.8377 |
| | Gemma Phr. | 0.7897 | 0.6058 | 0.1368 | 0.1915 | 0.0158 | 0.4005 | 0.8056 | 0.8908 | 0.8377 |
| | Llama Phr. | 0.7850 | 0.6045 | 0.1335 | 0.1950 | 0.0183 | 0.4198 | 0.8056 | 0.8908 | 0.8377 |
| | Original | 0.7962 | 0.6178 | 0.1623 | 0.2250 | 0.0380 | 0.4580 | 0.8056 | 0.8908 | 0.8377 |
| QwenVL | GPT-4o Phr. | 0.8347 | 0.6323 | 0.2250 | 0.2205 | 0.0483 | 0.4098 | 0.4375 | 0.7948 | 0.6793 |
| | Gemma Phr. | 0.8262 | 0.6278 | 0.2043 | 0.2025 | 0.0350 | 0.4160 | 0.4375 | 0.7926 | 0.6793 |
| | Llama Phr. | 0.8270 | 0.6303 | 0.2113 | 0.2213 | 0.0453 | 0.4470 | 0.4375 | 0.7948 | 0.6776 |
| | Original | 0.8427 | 0.6405 | 0.2548 | 0.2488 | 0.0798 | 0.4895 | 0.4375 | 0.7926 | 0.6776 |
| Llama 3.2 Vision | GPT-4o Phr. | 0.8663 | 0.6625 | 0.2663 | 0.2473 | 0.0795 | 0.6890 | 0.4667 | 0.9562 | 0.8937 |
| | Gemma Phr. | 0.8591 | 0.6575 | 0.2393 | 0.2315 | 0.0572 | 0.6810 | 0.4722 | 0.9562 | 0.8952 |
| | Llama Phr. | 0.8647 | 0.6613 | 0.2588 | 0.2545 | 0.0793 | 0.6915 | 0.4722 | 0.9546 | 0.8961 |
| | Original | 0.8704 | 0.6680 | 0.2990 | 0.2843 | 0.1060 | 0.6853 | 0.4722 | 0.9562 | 0.8937 |

We systematically paraphrased a representative subset of our ground truth captions (agriculture, crime, demography, Walmart) using architecturally distinct LLMs (GPT-4o, Gemma 27B, and Llama 90B), resulting in three additional sets of ground truth captions. The paraphrasing prompt was designed to instruct the model to thoroughly alter sentence structure, syntax, and word choice while preserving all factual information, numeric values, and statistical details with absolute fidelity. The prompt used for

paraphrasing is shown in N.3. We define this paraphrased caption set as CaTS-Bench-Paraphrased, which contains captions that are semantically equivalent to the original ground truth but differ only in linguistic style.

We then re-evaluated the outputs of all six representative pretrained models against CaTS-Bench-Paraphrased using our full suite of metrics. Results of this analysis are presented in the Table 10, one for each ground-truth generator's linguistic style. Values represent the average across selected domains. To ease comparison, we also report the results obtained with our original Gemini captions as ground truth.

Next, we provide an analysis on the rank correlation of model performances between the original and paraphrased evaluation settings. A high correlation in model rankings would indicate that our benchmark is robust to linguistic variation; the metrics would be consistently measuring the underlying semantic content and numeric accuracy of the captions, not their surface-level similarity to a specific writing style. A low correlation would suggest a non-trivial dependence on the oracle's particular linguistic patterns. For each linguistic evaluation metric, we provide the model ranking across the four linguistic styles in Figure 5.

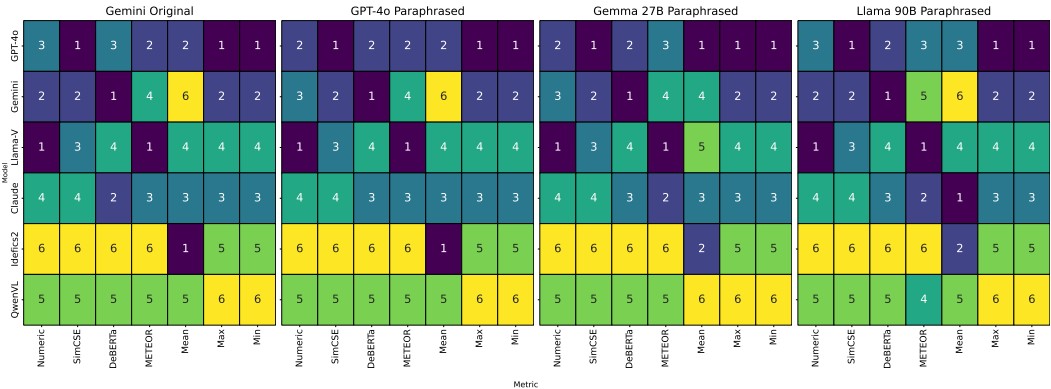

Figure 5: Model ranking heatmaps across metrics under four reference styles. Rankings: 1 (highest) to 6 (lowest). Model mappings: Gemini (Gemini 2.0 Flash), Claude (Claude 3 Haiku), Llama-V (Llama 3.2 Vision).

Qualitatively, we observe a negligible impact of model-specific linguistic style on the model rankings, suggesting that our evaluations are robust to particular linguistic styles. For each linguistic metric, we measure the average Spearman correlation between the ranking according to Gemini's style and the ranking according to the other three styles. See Table 11.

Table 11: Spearman correlation of model rankings of Gemini as ground truth vs. different models as ground truth. A Spearman Correlation of 1 means ranking does not change at all.

|  | DeBERTa F1 | SimCSE | BLEU | ROUGE-L | METEOR | **Average** |
|---|---|---|---|---|---|---|
| Spearman Correlation | 0.9714 | 1.0000 | 0.8138 | 0.9048 | 0.9429 | **0.9266** |
| p-value | 0.0018 | 0.0000 | 0.0557 | 0.0257 | 0.0079 | **0.0182** |

In summary, this experiment demonstrates that our evaluation framework is robust to variations in the linguistic style of the reference captions. This conclusion is quantitatively supported by the high Spearman correlation coefficients observed in model rankings between the original and paraphrased benchmark sets. These results indicate that **our evaluation framework captures the semantic fidelity and factual quality of the generated content, rather than rewarding models for merely mimicking the stylistic patterns of our oracle model, Gemini 2.0 Flash**. Consequently, the benchmark evaluates fundamental capabilities in time series understanding and description, not superficial stylistic alignment. Furthermore, Gemini 2.0 Flash maintains its superior rank in most metrics regardless of the linguistic style of the ground truth. This consistent dominance validates

its selection as a highly capable oracle, reinforcing that its utility stems from its intrinsic ability to generate high-quality descriptions rather than from any benchmark-specific bias.

## H.4 DIVERSITY ANALYSIS

### H.4.1 CONTENT IN THE CAPTIONS

We analyzed all the 4005 Gemini-generated test set captions using a structured keyword-based approach. Captions across eleven domains were scanned for statistical descriptors (e.g., *mean*, *average*, *standard deviation*, *maximum*, *minimum*, *range*) and trend-related expressions (e.g., *increase*, *decrease*, *stability*, *volatility*, *seasonality*). These terms were grouped into clusters such as *Central Tendency*, *Dispersion*, *Extremes*, *Increasing/Decreasing Trends*, *Stability and Volatility*, and *Comparative Trends*. The results in Table 12 show that **captions consistently draw from a diverse mix of descriptors, spanning both statistical features and temporal patterns**. While some categories (e.g., percentiles, distribution-shape terms) were rare, coverage of core descriptors was broad, and every caption included at least one statistical or trend-related element.

Table 12: Coverage of statistical and trend descriptor clusters across benchmark captions. Captions consistently include diverse descriptors capturing both quantitative and temporal aspects of the data.

| Category | Cluster and Keywords in the cluster | Captions |
|---|---|---|
| Statistical | Central Tendency (mean, average, median, mode) | 3930 |
| | Minimum Values (min, minimum, lowest value) | 1528 |
| | Maximum Values (max, maximum, highest value) | 1487 |
| | Dispersion (std, variance, deviation, iqr) | 1376 |
| | Range/Spread (range, spread) | 492 |
| | Extremes (peak, spike, dip, trough) | 2966 |
| Trend and Patterns | Peaks and Valleys (peak, dip, spike, trough) | 3013 |
| | Increasing Trends (increase, rising, growing) | 1987 |
| | Decreasing Trends (decrease, drop, falling) | 2197 |
| | Comparative Trends (higher/lower, difference) | 2217 |
| | Stability and Volatility (stable, fluctuating) | 2276 |

### H.4.2 N-GRAM DIVERSITY

We measured type token ratios (TTR) across $411k$ tokens in the semi-synthetic test set. The ratios rise significantly with $n$: TTR = 0.0288 (1 gram), 0.1288 (2 gram), 0.2971 (3 gram), 0.4638 (4 gram), and 0.6050 (5 gram), indicating that the phrases diversify rapidly as $n$ increases. Notably, we observe over $250k$ unique 5-grams out of $411k$ tokens, providing strong evidence that the captions are not overly templated.

### H.4.3 LATENT SIMILARITIES

We further assessed whether Gemini-generated captions introduce systematic stylistic or linguistic bias. Results across multiple embedding models show minimal template reliance and high diversity. We performed $\sim 8M$ pairwise comparisons across 4005 captions using nine embedding models (Table 13). Intra-domain similarity was consistently higher ($0.59 - 0.78$) than inter-domain similarity ($0.23 - 0.54$), with large effect sizes (Cohen's $d > 3.26$). Near-duplicates (cosine $> 0.95$) were rare, with an average of $2.3\%$ of pairs. Even within domains, similarity showed non-trivial variance, indicating that **captions are not rigid templates but semantically varied**.

Table 13: Embedding similarities

| Encoder Model | Dimensionality | Mean | Median | STD | Intra | Inter | % Pairs $> 0.95$ |
|---|---|---|---|---|---|---|---|
| MiniLM-L12-v2 | 368 | 0.2932 | 0.2500 | 0.1771 | 0.6202 | 0.2269 | 5.26 |
| MiniLM-L6-v2 | 384 | 0.3339 | 0.3013 | 0.1510 | 0.5999 | 0.2800 | 1.47 |
| mpnet-base-v2 | 768 | 0.3278 | 0.2791 | 0.1775 | 0.6631 | 0.2599 | 8.29 |
| bge-large-v1.5 | 1024 | 0.5807 | 0.5605 | 0.0986 | 0.7600 | 0.5443 | 1.67 |
| mxbai-large-v1 | 1024 | 0.5293 | 0.5032 | 0.1116 | 0.7370 | 0.4871 | 2.15 |
| Qwen3-4B | 2560 | 0.3816 | 0.3530 | 0.1273 | 0.6098 | 0.3354 | 0.20 |
| e5-mistral-7b | 4096 | 0.6507 | 0.6358 | 0.0697 | 0.7770 | 0.6251 | 1.33 |
| GritLM-7B | 4096 | 0.4767 | 0.4537 | 0.0947 | 0.6481 | 0.4420 | 0.04 |
| Qwen3-8B | 4096 | 0.3736 | 0.3465 | 0.1225 | 0.5947 | 0.3288 | 0.32 |
| **Average** | - | **0.4386** | **0.4092** | **0.1256** | **0.6678** | **0.3922** | **2.30** |

### H.4.4 HUMAN VS. AI EMBEDDING SIMILARITY

We computed cosine similarities between captions authored by humans, GPT-4o, and Gemini using Qwen-8B embeddings. Both small-scale (22 vs. 22 captions) and larger-scale comparisons (22 vs. 150 captions; $3k$ vs. $3k$ STOCK dataset (Jhamtani & Berg-Kirkpatrick, 2021)) show only modest gaps between human–AI and human–human similarity. For random CaTS-Bench samples, we report averages across five runs with deviations in the range 0.0010–0.0174. See Table 14 for full results.

Table 14: Cosine similarities between human, GPT-4o, and Gemini captions. **Human–AI similarity closely tracks human–human similarity, suggesting minimal stylistic divergence**.

| | Balanced (22 vs 22) | | | Unbalanced (22 vs 150) | | | STOCK (3k vs 3k) | | |
|---|---|---|---|---|---|---|---|---|---|
| **Comparison** | Human | GPT | Gemini | Human | GPT | Gemini | Human | GPT | Gemini |
| Human | 0.371 | 0.341 | 0.353 | 0.371 | 0.358 | 0.353 | 0.654 | 0.662 | 0.586 |
| GPT | | 0.406 | 0.328 | | 0.401 | 0.329 | | 0.694 | 0.604 |
| Gemini | | | 0.370 | | | 0.378 | | | 0.599 |

### H.5 REPRODUCIBILITY VALIDATION

We re-ran our evaluation three times on $\sim$500 randomly sampled test examples from the semi-synthetic set across five representative models (GPT-4o, Claude 3 Haiku, LLaMA, Idefics, Qwen-VL). Figure 6 shows a log-scale visualization. **Across nearly all metrics, the variance is vanishingly small, often on the order of $10^{-6}$, which confirms stability and supports single-run robustness**.

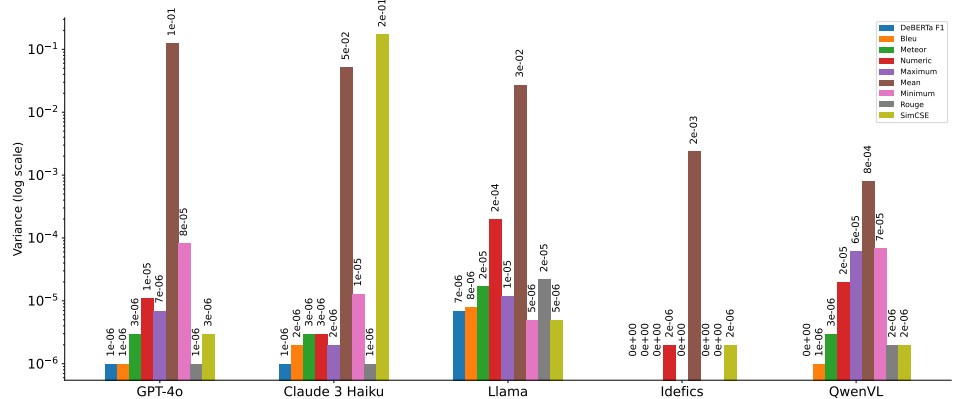

Figure 6: Variance across three independent runs (approximately 500 samples) for each model–metric pair. The logarithmic scale highlights both very small variances ($10^{-6}$) and moderately larger values.

# I  ROLE OF VISION

## I.1  VISUAL MODALITY ABLATION

We present Table 15, comparing the performance of VLMs with and without the visual input.

Table 15: Evaluation of generated semi-synthetic captions under modality ablation. Each metric is split into two columns: **VL** (vision-language input) and **L** (text-only input).

| Model | DeBERTa F1 | | SimCSE | | BLEU | | ROUGE-L | | METEOR | | Numeric | |
| --- | --- | --- | --- | --- | --- | --- | --- | --- | --- | --- | --- | --- |
| | VL | L | VL | L | VL | L | VL | L | VL | L | VL | L |
| InternVL | 0.659 | 0.677 | 0.794 | 0.854 | 0.081 | 0.113 | 0.247 | 0.283 | 0.260 | 0.317 | 0.589 | 0.636 |
| QwenVL | 0.643 | 0.648 | 0.790 | 0.802 | 0.081 | 0.090 | 0.249 | 0.258 | 0.260 | 0.274 | 0.503 | 0.520 |
| Phi-4 M.I. | 0.624 | 0.674 | 0.797 | 0.845 | 0.074 | 0.118 | 0.274 | 0.290 | 0.239 | 0.306 | 0.583 | 0.634 |
| SmolVLM | 0.594 | 0.603 | 0.692 | 0.758 | 0.043 | 0.068 | 0.224 | 0.244 | 0.178 | 0.257 | 0.473 | 0.565 |
| Llama 3.2 Vision | 0.670 | 0.669 | 0.850 | 0.849 | 0.117 | 0.110 | 0.290 | 0.275 | 0.314 | 0.313 | 0.684 | 0.598 |
| Idefics2-8B | 0.604 | 0.632 | 0.698 | 0.816 | 0.040 | 0.080 | 0.225 | 0.258 | 0.161 | 0.270 | 0.454 | 0.585 |
| LLaVA v1.6 Mistral | 0.650 | 0.648 | 0.820 | 0.824 | 0.086 | 0.098 | 0.259 | 0.261 | 0.287 | 0.286 | 0.517 | 0.532 |
| Claude 3 Haiku | 0.682 | 0.676 | 0.856 | 0.853 | 0.112 | 0.117 | 0.291 | 0.287 | 0.300 | 0.298 | 0.628 | 0.609 |
| Gemini 2.0 Flash | 0.688 | 0.698 | 0.858 | 0.871 | 0.137 | 0.175 | 0.318 | 0.343 | 0.279 | 0.328 | 0.677 | 0.684 |

## I.2  VISUAL ATTENTION ANALYSIS

Interpreting visual grounding in large multimodal models is non-trivial, as not all of them expose interpretable cross-modal attention mechanisms. We attempt this using the LLaVA model, which provides access to decoder-level cross-attention weights over vision tokens. We adapt the approach in Zhang (2024) for the `LLaVA 1.6` model. We visualize per-token visual grounding via the following steps. For each generated token, we extract the decoder cross-attention matrix $\mathbf{A}_{\text{llm}} \in \mathbb{R}^{T \times V}$, where $T$ is the number of generated tokens and $V$ is the number of vision tokens.

Next, we zero out the attention to the beginning-of-sequence token and normalize each row:

$$\tilde{\mathbf{A}}_{\text{llm}}[t, v] = \begin{cases} 0, & \text{if } v = \texttt{<bos>} \\ \frac{\mathbf{A}_{\text{llm}}[t,v]}{\sum_{v'} \mathbf{A}_{\text{llm}}[t,v']}, & \text{otherwise} \end{cases} \tag{6}$$

From the CLIP style vision encoder, we extract attention matrices $\mathbf{A}_{\text{vit}}^{(l)} \in \mathbb{R}^{V \times V}$ from multiple layers and average them:

$$\bar{\mathbf{A}}_{\text{vit}} = \frac{1}{L} \sum_{l=1}^{L} \mathbf{A}_{\text{vit}}^{(l)} \tag{7}$$

For each token $t$, we compute its attention-weighted vision token distribution and project it back to the 2D image grid, and the projected map $\hat{\mathbf{H}}_t$ is rendered as a heatmap and overlaid on the original image. This allows inspection of which visual regions contribute to each generated token.

$$\hat{\mathbf{H}}_t = \texttt{Upsample}\left(\text{reshape}(\mathbf{H}_t, \text{grid})\right) \tag{8}$$

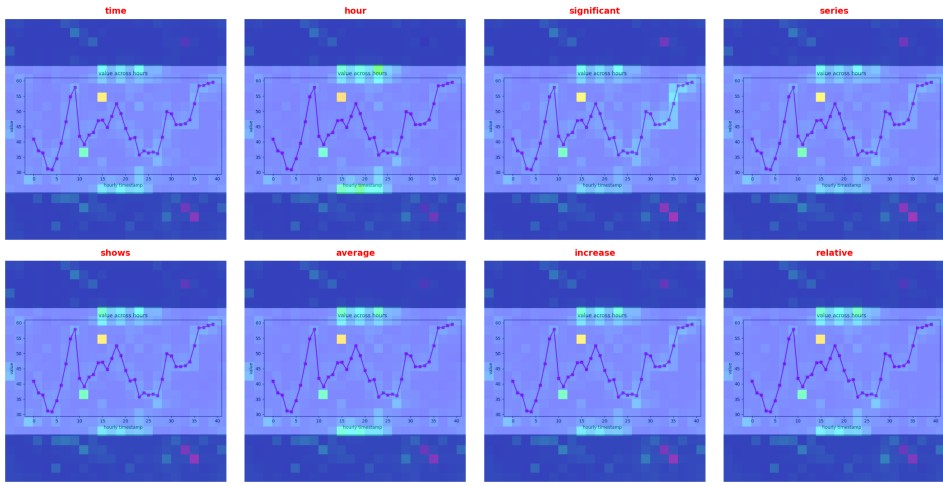

Figure 7: Word-level attention maps for the top 8 tokens from `LLaVA 1.6 Mistral` overlaid on a time series plot. Despite expectations of alignment with visual trends, attention remains largely diffuse, offering only weak evidence of localized visual grounding in caption generation.

## I.3 TESTING ALTERNATIVE VISUAL ENCODINGS

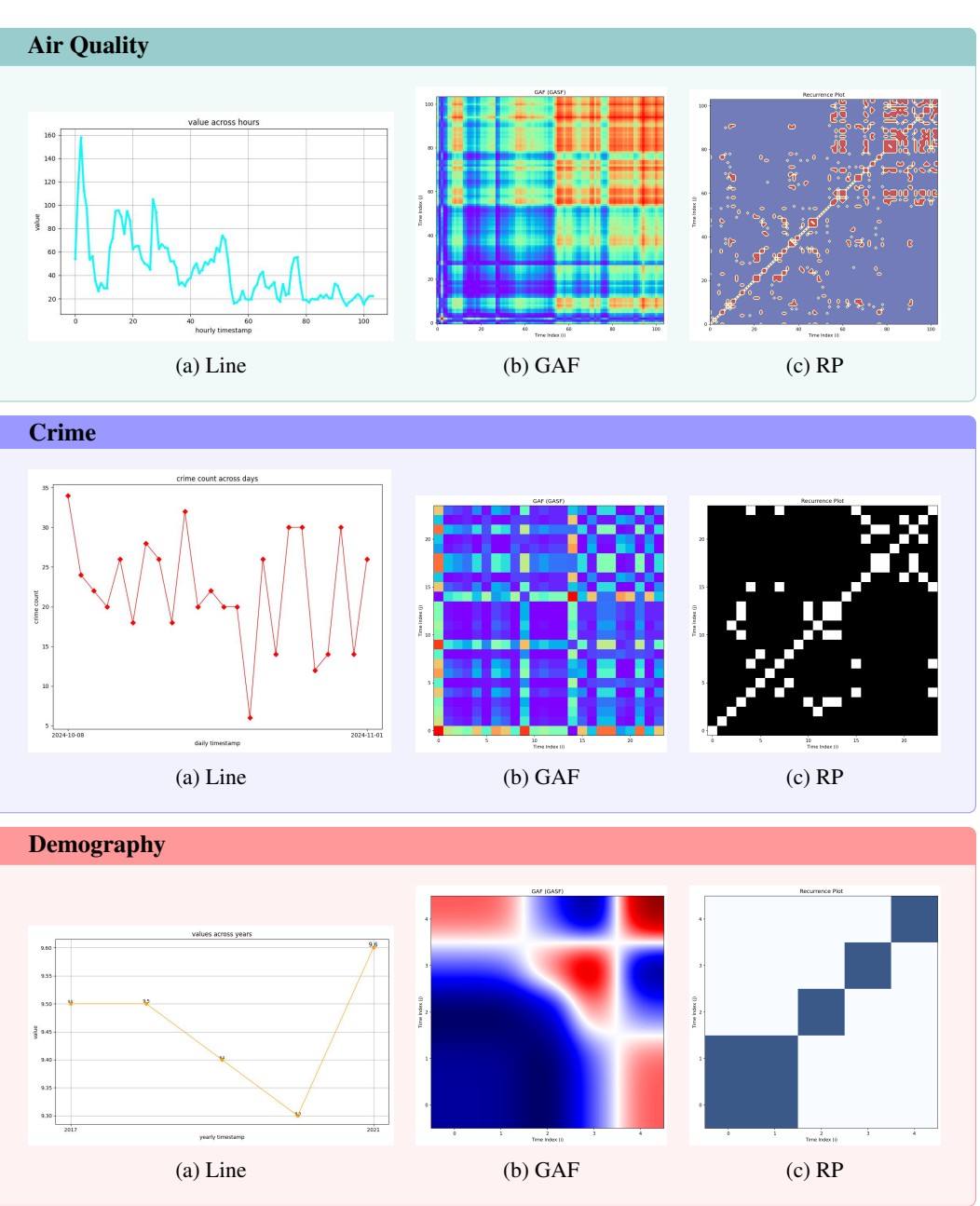

Figure 8: Examples of visual encodings across three domains across three sample domain (Air Quality, Crime, Demography) (a) Line, (b) GAF, and (c) RP.

To investigate whether more complex visualizations could help in the modality collapse issue, we experimented with Gramian Angular Fields (GAFs) and recurrence plots (RPs) in addition to standard line plots. Each was generated from the same univariate time series windows, using first-order deltas as input. There are also other encodings, such as multi-series overlays or confidence intervals, but we find these are less applicable in our strictly univariate setting.

Table 16: Effect of alternative visual encodings on captioning performance on subset of semi-synthetic captions. Values show the baseline score for *No Plot (TS+Text)* and relative differences (Δ) for line plots, Gramian Angular Fields (GAFs), and recurrence plots (RPs).

| Model | Metric | No Plot | Line (Δ) | GAF (Δ) | RP (Δ) |
|---|---|---|---|---|---|
| Idefics2 (8B) | DeBERTa F1 | 0.6255 | -0.078 | -0.0525 | -0.0525 |
| | BLEU | 0.069 | -0.0405 | -0.0460 | -0.0430 |
| | METEOR | 0.2275 | -0.0925 | -0.0905 | -0.0815 |
| | NUMERIC | 0.4455 | -0.1665 | -0.1715 | -0.1285 |
| | MAXIMUM | 0.8080 | +0.0730 | -0.1520 | -0.1300 |
| | MEAN | 0.4195 | +0.1285 | +0.0945 | +0.2995 |
| | MINIMUM | 0.7590 | -0.0845 | -0.2510 | -0.3940 |
| | STD | 0.5500 | -0.1835 | -0.1060 | -0.3360 |
| | ROUGE | 0.2375 | -0.0460 | -0.0465 | -0.0415 |
| | SIMCSE | 0.7915 | -0.1735 | -0.1935 | -0.1705 |
| LLaMA 3.2 Vision | DeBERTa F1 | 0.6550 | -0.0020 | -0.0030 | -0.0080 |
| | BLEU | 0.0840 | +0.0005 | -0.0090 | -0.0100 |
| | METEOR | 0.2865 | -0.0080 | -0.0195 | -0.0315 |
| | NUMERIC | 0.5070 | +0.0315 | +0.0120 | +0.0170 |
| | MAXIMUM | 0.8611 | +0.0584 | -0.0381 | -0.0151 |
| | MEAN | 0.5395 | +0.0690 | +0.0755 | +0.1215 |
| | MINIMUM | 0.6670 | +0.0820 | +0.0740 | +0.0670 |
| | STD | 0.8750 | -0.3270 | -0.2310 | -0.1250 |
| | ROUGE | 0.2570 | +0.0040 | 0.0000 | -0.0040 |
| | SIMCSE | 0.8265 | -0.0020 | -0.0085 | -0.0235 |
| Qwen-VL | DeBERTa F1 | 0.6315 | -0.0045 | -0.0045 | -0.0095 |
| | BLEU | 0.0645 | -0.0040 | +0.0025 | -0.0015 |
| | METEOR | 0.2385 | -0.0050 | +0.0025 | -0.0025 |
| | NUMERIC | 0.3555 | +0.0560 | -0.0675 | -0.0515 |
| | MAXIMUM | 0.6610 | +0.1135 | +0.0200 | +0.0430 |
| | MEAN | 0.4510 | -0.0110 | -0.2150 | -0.2570 |
| | MINIMUM | 0.5350 | -0.0055 | -0.0650 | -0.0620 |
| | STD | 0.1070 | +0.2095 | -0.0620 | -0.0070 |
| | ROUGE | 0.2310 | -0.0045 | +0.0130 | +0.0030 |
| | SIMCSE | 0.7835 | +0.0020 | -0.0085 | -0.0090 |

Table 16 reports results for three representative models (Idefics2-8B, LLaMA 3.2 Vision, Qwen-VL). Values show the baseline metric with no plot (TS+Text only), followed by relative gains or losses for each visualization type. Across models and metrics, neither GAFs nor recurrence plots significantly improved performance; in many cases, they degraded results relative to line plots or even no visual input. These findings suggest that the bottleneck lies not in the choice of visualization but in the models' inability to effectively integrate visual cues. With this, we motivate the development of models and encodings that better exploit the structured information available in visualized time series.

## J Q&A Tasks

We present accuracy scores for VLMs on the Q&A task in Table 17. An analysis of the highlighted statistics reveals a striking contrast between the finetuned and pretrained models. The finetuned model frequently produces highly confident yet sometimes incorrect predictions, whereas the pretrained model demonstrates more caution, acknowledging that the mean is lower than expected without attempting to estimate a specific value. Notably, certain proprietary models are now reaching, and at times even surpassing, human performance on specific subsets of tasks. While this signals exciting progress in the field, it also highlights the nuances of human cognitive performance, particularly under conditions where distraction might occur. It is vital to note, however, that no singular model has consistently achieved near-human proficiency across the entirety of the benchmark's demands. The plot retrieval task, in particular, stands out as a significant hurdle, robustly affirming the unparalleled human capacity for holistic visual-numeric integration, a critical frontier for time series understanding.

Table 17: Model accuracy for time-series Q&A tasks. **Bolded** and underlined scores respectively denote first and second places (excluding human performance). Caption/Plot/TS refer to caption, plot, and time series matching. Amplitude/Peak Earlier/Mean/Variance refer to amplitude, peak, mean, and variance comparison. Green and Red indicate improvement and degradation after finetuning, respectively.

| Model | Caption | Plot | TS | Amplitude | Peak Earlier | Mean | Variance |
|---|---|---|---|---|---|---|---|
| **Proprietary** | | | | | | | |
| Gemini 2.0 Flash | 0.78 | 0.30 | 0.61 | 0.8 | 0.42 | 0.7 | 0.62 |
| Gemini 2.5 Pro Preview | 0.66 | 0.30 | 0.31 | **1.0** | **1.0** | **1.0** | **0.85** |
| Claude 3 Haiku | 0.68 | 0.29 | 0.57 | 0.65 | 0.40 | 0.53 | 0.33 |
| GPT-4o | **0.96** | 0.31 | **0.77** | 0.825 | 0.725 | 0.7 | 0.625 |
| **Pretrained** | | | | | | | |
| InternVL 2.5 | 0.55 | 0.17 | 0.49 | 0.60 | 0.47 | 0.45 | 0.40 |
| LLaVA v1.6 Mistral | 0.39 | 0.27 | 0.32 | 0.45 | 0.45 | 0.42 | 0.45 |
| Phi-4 M.I. | 0.62 | 0.29 | 0.45 | 0.7 | 0.82 | 0.68 | 0.7 |
| Idefics 2 | 0.49 | 0.25 | 0.29 | 0.35 | 0.4 | 0.4 | 0.5 |
| SmolVLM | 0.26 | **0.34** | 0.28 | 0.4 | 0.48 | 0.44 | 0.6 |
| QwenVL | 0.68 | 0.27 | 0.61 | 0.7 | 0.5 | 0.6 | 0.4 |
| Llama 3.2 Vision | 0.66 | 0.24 | 0.27 | 0.45 | 0.63 | 0.43 | 0.3 |
| **Finetuned** | | | | | | | |
| LLaVA v1.6 Mistral | 0.44 | 0.25 | 0.29 | 0.43 | 0.53 | 0.35 | 0.4 |
| Phi-4 M.I. | 0.59 | 0.29 | 0.45 | 0.83 | 0.88 | 0.7 | 0.55 |
| Idefics 2 | 0.33 | 0.23 | 0.29 | 0.58 | 0.38 | 0.5 | 0.63 |
| SmolVLM | 0.18 | 0.26 | 0.29 | 0.28 | 0.48 | 0.38 | 0.58 |
| QwenVL | 0.55 | 0.25 | 0.43 | 0.7 | 0.4 | 0.58 | 0.58 |
| Llama 3.2 Vision | 0.66 | 0.24 | 0.27 | 0.4 | 0.6 | 0.43 | 0.33 |
| *Human* | 0.81 | 0.95 | 0.83 | 0.925 | 0.85 | 0.95 | 0.90 |

## J.1 SAMPLE Q&A QUESTIONS

We provide examples of Q&A questions in Figures 9, 10, 12, 11, 13, 14, and 15, covering one example per question type.

---

**Question**

Given two time series A and B, detect which one has a higher amplitude defined as maximum - minimum.

A: [1.15, 0.92, 0.85, 0.75, 0.57, 0.62, 0.6, 0.5, 0.68, 0.72, 0.8, 0.67, 0.8, 0.55, 0.55, 0.7, 0.88]

B: [87.0, 83.0, 77.0, 74.0, 84.0]

You must respond only with valid JSON, and no extra text or markdown.

The JSON schema is:
`{"answer": <string>}`
`<string>` must be an answer string containing only A, B.
Ensure your output parses as JSON with exactly one top-level object containing the answer field.

**Answer**
`"answer": "B"`

---

Figure 9: Example of a *time series amplitude comparison* question.

**Question**

Given two time series A and B, detect which one reaches its maximum earlier.

A: [66.76, 83.06, 85.77, 90.77, 98.81, 90.62, 80.05, 91.36, 89.59, 76.4, 80.1, 85.6, 84.41]

B: [949.0, 689.0, 561.0, 552.0, 563.0]

You must respond only with valid JSON, and no extra text or markdown.

The JSON schema is:
`{"answer":  <string>}`
`<string>` must be an answer string containing only A, B.
Ensure your output parses as JSON with exactly one top-level object containing the answer field.

**Answer**
`"answer":  "B"`

Figure 10: Example of a *time series peak comparison* question.

**Question**

Given the following two time series A and B, please identify which one has higher volatility.

A: [0.14, 0.14, 0.14, 0.29, 0.29, 0.29, 0.29, 0.29, 0.29, 0.29, 0.57, 0.57, 0.57, 0.57, 0.57, 0.57]

B: [0.21, 0.33, 0.41, 0.39, 0.44, 0.35, 0.35, 0.43, 0.51, 0.65, 0.69, 0.74]

You must respond only with valid JSON, and no extra text or markdown.

The JSON schema is:
`{"answer":  <string>}`
`<string>` must be an answer string containing only A, B.
Ensure your output parses as JSON with exactly one top-level object containing the answer field.

**Answer**
`"answer":  "A"`

Figure 11: Example of a *time series variance comparison* question.

**Question**

Given the following two time series A and B, please identify which one has higher overall values.

A: [65.0, 65.0, 64.0, 37.0, 55.0, 51.0]

B: [6.29, 6.29, 6.29, 7.0, 7.0, 7.0, 7.0, 6.71, 6.71, 6.71, 6.71, 6.717, 7.57, 7.57, 7.14, 7.14, 7.14, 7.14, 7.43]

You must respond only with valid JSON, and no extra text or markdown.

The JSON schema is:
`{"answer":  <string>}`
`<string>` must be an answer string containing only A, B.
Ensure your output parses as JSON with exactly one top-level object containing the answer field.

**Answer**
`"answer":  "A"`

Figure 12: Example of a *time series mean comparison* question.

**Question**

Here is a time series:
37.00,37.00,37.00,37.00,37.00,40.57,40.57,40.57,40.57,40.57,40.57

What caption best describes this time series?

(A) From May 1st to July 26th, 2024, the daily COVID-19 deaths in China show a fluctuating pattern, with values generally ranging between 0.29 and 2.86. There are periods of relative stability, such as the initial days of May with a consistent 0.86, interspersed with occasional spikes to 2.86, and dips to 0.29 towards the end of July. Compared to the general daily death statistics for China, where the mean is 73.0 and the maximum reaches 6812.0, this specific time series indicates a period of significantly lower daily deaths, suggesting a substantial improvement in the COVID-19 situation during this timeframe.

(B) From October 24, 2023, to November 3, 2023, the daily COVID-19 cases in Luxembourg show a relatively stable pattern, beginning at 37.3 cases and rising to 40.57 cases by October 29, 2023, where it remains for the rest of the period. Compared to the country's general statistics, where the mean is 236, the daily cases during this period are significantly lower, suggesting a period of reduced viral transmission. This trend does not follow any expected seasonal patterns, as COVID-19 case numbers are known to fluctuate unpredictably.

(C) From October 24, 2023, to November 3, 2023, the daily COVID-19 cases in Luxembourg show a relatively stable pattern, beginning at 37 cases and rising to 40.57 cases by October 29, 2023, where it remains for the rest of the period. Compared to the country's general statistics, where the mean is 236.0, the daily cases during this period are significantly lower, suggesting a period of reduced viral transmission. This trend does not follow any expected seasonal patterns, as COVID-19 case numbers are known to fluctuate unpredictably.

(D) From October 24, 2023, to November 3, 2023, the daily COVID-19 cases in Luxembourg show a relatively unstable pattern, beginning at 37 cases and decreasing to 40.57 cases by October 29, 2023, where it remains for the rest of the period. Compared to the country's general statistics, where the mean is 236.0, the daily cases during this period are significantly lower, suggesting a period of reduced viral transmission. This trend does follow expected seasonal patterns, as COVID-19 case numbers are known to fluctuate unpredictably.

You must respond only with valid JSON, and no extra text or markdown.

The JSON schema is:
{"answer": <string>}
<string> must be an answer string containing only A, B, C, or D.
Ensure your output parses as JSON with exactly one top-level object containing the answer field.

**Answer**

```
"answer": "C"
```

Figure 13: Example of a *caption matching* question.

**Question**

Here is a time series:
186.57, 186.57, 186.57, 186.57, 186.57, 150.29, 150.29, 150.29, 150.29, 150.29, 150.29, 150.29, 103.14, 103.14, 103.14, 103.14, 103.14, 103.14, 103.14, 77.00, 77.00, 77.00, 77.00, 77.00, 77.00, 77.00, 52.71, 52.71, 52.71, 52.71, 52.71, 52.71, 52.71, 41.71, 41.71, 41.71, 41.71, 41.71, 41.71, 41.71, 39.71, 39.71, 39.71, 39.71, 39.71, 39.71, 39.71, 29.86, 29.86, 29.86, 29.86, 29.86, 29.86, 29.86, 27.43, 27.43, 27.43, 27.43, 27.43, 27.43, 27.43, 22.57, 22.57, 22.57, 22.57, 22.57, 22.57, 22.57, 15.14, 15.14, 15.14, 15.14, 15.14, 15.14, 15.14, 18.71, 18.71, 18.71, 18.71, 18.71, 18.71, 22.71, 22.71, 22.71, 22.71, 22.71, 22.71, 22.71, 23.14, 23.14, 23.14, 23.14, 23.14, 23.14, 21.00, 21.00, 21.00, 21.00, 21.00, 21.00, 21.00, 30.57, 30.57, 30.57, 30.57, 30.57, 30.57, 30.57, 30.57, 30.57, 30.57, 30.57, 30.57, 30.57, 36.29, 36.29, 36.29, 36.29, 36.29, 36.29, 36.29, 59.71, 59.71, 59.71, 59.71, 59.71, 59.71, 59.71, 93.71, 93.71, 93.71, 93.71, 93.71, 93.71, 93.71, 140.86, 140.86, 140.86, 140.86, 140.86, 140.86, 140.86

Here are four plots of different time series:

(A)

(B)

(C)

(D)

Which plot corresponds to the time series provided above?

You must respond only with valid JSON, and no extra text or markdown.

The JSON schema is:
`{"answer": <string>}`
`<string>` must be an answer string containing only A, B, C, or D.
Ensure your output parses as JSON with exactly one top-level object containing the answer field.

**Answer**

`"answer": "A"`

Figure 14: Example of a *plot matching* question.

> **Question**
>
> Here is a time series caption:
> From 2014 to 2019, Bulgaria's Agricultural output index (2015=100) generally increased, starting at 103.4 in 2014 and reaching a peak of 109.23 in 2019, with a slight dip to 100.0 in 2015. The average output index during this period was 105.4, notably lower than the historical mean of 126.73, suggesting a period of relatively lower agricultural productivity compared to Bulgaria's long-term performance. The increase from 2015 to 2019 indicates a moderate recovery and growth phase-t within this specific timeframe.
>
> What time series is best described by this caption?
> (A) [109.23, 107.24, 108.45, 104.1, 100.0, 103.4]
> (B) [108.45, 100.0, 104.1, 107.24, 103.4, 109.23]
> (C) [103.9, 99.8, 104.1, 109.2, 106.8, 109.23]
> (D) [103.4, 100.0, 104.1, 108.45, 107.24, 109.23]
>
> You must respond only with valid JSON, and no extra text or markdown.
>
> The JSON schema is:
> {"answer": <string>}
> <string> must be an answer string containing only A, B, C, or D.
> Ensure your output parses as JSON with exactly one top-level object containing the answer field.

> **Answer**
> "answer": "D"

Figure 15: Example of a *time series matching* question.

## J.2 QWEN-BASED FILTERING

To show that questions erroneously answered by `Qwen 2.5 Omni` are indeed harder, we evaluated a subset of models on both an easy set of 600 questions and the hard set generated by `Qwen 2.5 Omni`. The questions in the easy set are randomly sampled from those correctly answered by `Qwen 2.5 Omni`. Table 18 depicts the comparison. All models, regardless of architecture, show a consistent performance gain on the "easy" subset, demonstrating that Qwen-filtered questions are broadly harder, not uniquely harder for Qwen. To ensure a balanced benchmark, we release both the full set (38.4k questions) and the hard subset (7k questions), enabling evaluation and training across the entire difficulty spectrum.

Table 18: Performance on easy vs. hard questions and corresponding lift.

| Model | Easy | Hard | Lift |
|---|---|---|---|
| Idefics 2 | 65% | 46% | +19% |
| InternVL 2.5 | 72% | 43% | +29% |
| Phi-4 | 61% | 46% | +15% |
| SmolVLM | 53% | 48% | +5% |
| Llama-3.2 | 54% | 49% | +5% |

## J.3 DISTRACTOR GENERATION IN Q&A TASKS

To increase task difficulty, artificial perturbations are applied in the Time Series Matching and Caption Matching tasks. As shown in the table below, these perturbations significantly impacted model performance in Time Series Matching, increasing task difficulty and forcing models to reason over trends rather than relying on superficial cues.

To illustrate the rationale of this design choice, assume the correct time series option is [1,2,3]. Having distractors like [101, 99, 102] makes the question trivial due to its totally different scale and nature. Our distractors are generated by perturbations on the correct time series, resulting in the following distractors.

1. **Shuffled**: [2,1,3], avoids answering correctly by exploiting numeric lookup without temporal order awareness.

2. **Reversed**: [3,2,1], avoids reasoning without trend awareness.

3. **Gaussian-noised**: `[1.03, 1.99, 3.002]`, forces precise numeric reasoning instead of superficial numeric and trend similarity.

Table 19: Qwen-2.5-Omni-7B accuracy by task and distractor type.

| Question Type | Distractor Type | Accuracy |
|---|---|---|
| Time Series Matching | Cross-domain | 0.9803 |
| Time Series Matching | Same-domain | 0.9586 |
| Time Series Matching | Artificially Perturbed | **0.6864** |
| Caption Matching | Cross-domain | 0.8325 |
| Caption Matching | Same-domain | **0.8250** |
| Caption Matching | Artificially Perturbed | 0.8532 |

Table 19 shows that these perturbations indeed augment question difficulty. For Caption Matching, we do not find a significant difference between artificially perturbed distractors compared to original negative examples drawn from the dataset. We release the set of Q&A questions without artificial perturbations as well. Perturbations are not applied to the other tasks; negative samples are instead drawn directly from other time series in the dataset.

## K  INFERENCE EXAMPLES

### K.1  NUMERIC TIME SERIES ABLATION

We compare the generated captions with and without explicitly providing the numeric time series data as input to the model in the prompt. As seen in Figure 16, when a numeric time series is included, the model produces some factual and interpretative errors. Notably, it describes the trend as "increased steadily," despite the clear dip in 2014 and a decline post-2017. It also incorrectly identifies 2018 as the year of the peak value 102.48, while the actual peak occurs in 2017. Similarly, the slight dip is misattributed to 2016 instead of the correct year 2014. In contrast, when the numeric time series is removed from the input, the generated caption becomes significantly more erroneous. The model fabricates plausible-sounding but incorrect values, for example, claiming the index reached 90.5 in 2009 and spiked to 105 in 2014, neither of which is present in the actual plot. This suggests that the absence of explicit numeric context forces the model to hal-

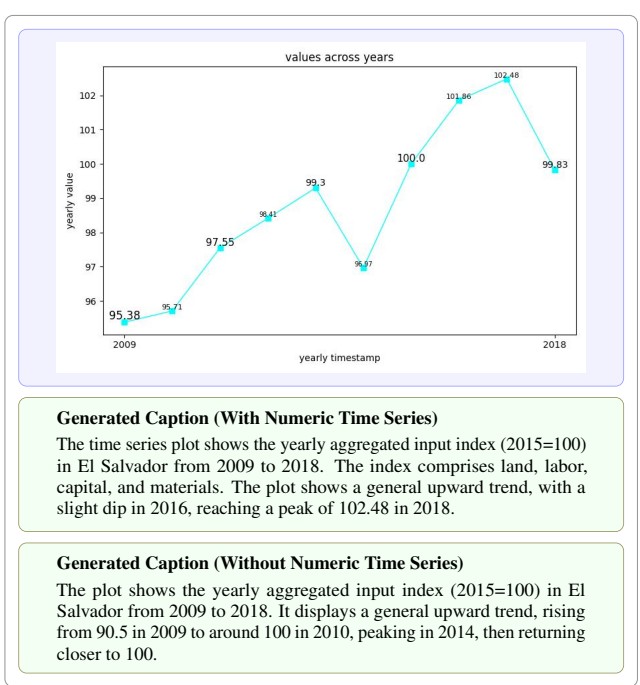

**Generated Caption (With Numeric Time Series)**

The time series plot shows the yearly aggregated input index (2015=100) in El Salvador from 2009 to 2018. The index comprises land, labor, capital, and materials. The plot shows a general upward trend, with a slight dip in 2016, reaching a peak of 102.48 in 2018.

**Generated Caption (Without Numeric Time Series)**

The plot shows the yearly aggregated input index (2015=100) in El Salvador from 2009 to 2018. It displays a general upward trend, rising from 90.5 in 2009 to around 100 in 2010, peaking in 2014, then returning closer to 100.

Figure 16: Comparison of generated captions with and without numeric time series input. Model evaluated is `LLaVA 1.6`.

lucinate plausible trajectories based solely on the shape of the line plot. While both versions demonstrate limitations in temporal precision, the numeric-aware caption is more grounded and less prone to hallucinating specific values, producing some factual errors but still outperforming the numeric-agnostic version, which fabricates values entirely.

## K.2   Statistical Inference Failure & Success

In the main paper, we mentioned that finetuned models often become overconfident when inferring statistical properties such as means and standard deviations, despite lacking the capability to compute them accurately. In this section, we present two concrete cases that illustrate this overconfidence in practice, and one case where the finetuned model actually successfully inferred the statistics.

**Case 1: Failure**   The following error case shows the finetuned `LLaVA v1.6 Mistral` erroneously attempting to infer the time series mean, while its pretrained version does not. We compare the captions generated by both versions of the model against the numericly grounded reference.

---

**Ground-Truth Caption**

The Total Factor Productivity (TFP) index for Malawi from 2008 to 2017 reveals notable fluctuations, starting at 85.99 in 2008 and peaking at 110.47 in 2017, with 2015 set at 100.0. There's an initial rise from 2008 to 2009, followed by some volatility before a significant jump to 107.82 in 2013, indicating a period of strong productivity growth, and the mean of **98.93** for this period is significantly higher than the historical mean of **61.88**, suggesting a period of above-average productivity compared to Malawi's longer-term performance.

**Caption from Pretrained Model**

The time series for Malawi's Total Factor Productivity index shows a generally upward trend from 2008 to 2017, with some fluctuations. The index starts at 85.99 in 2008 and reaches a peak of 110.47 in 2014. It then experiences a decline, dropping below 100 in 2015 and 2016. In 2017, the index showed a slight increase, reaching 100.06. Compared to global or regional norms, Malawi's productivity index appears to be lower than expected, suggesting a need for economic development and productivity enhancement strategies.

**Caption from Finetuned Model**

From 2008 to 2017, Malawi's Total Factor Productivity index (2015=100) exhibited considerable volatility, starting at 85.99 in 2008 and ending at 110.47 in 2017. The index experienced a notable increase from 2008 to 2010, rising from 85.99 to 95.76, followed by a dip to 91.75 in 2011, before generally trending upwards with some fluctuations. Compared to the historical mean of **94.84**, the mean of this specific time series is moderately higher at **99.88**, suggesting a period of relatively stronger productivity growth than the broader historical context.

---

Figure 17: Finetuned model attempts to infer the mean, but it fails. Compare the **bolden** values.

We observe that, while the pretrained model refrains from inferring the mean of the time series, displaying a form of "self-awareness" about its limitations, the finetuned model tends to make confident guesses. This behavior underscores a key limitation of the cross-entropy loss objective. Since ground-truth captions often include the time series mean, the model can reduce its loss by always attempting to predict them, as number tokens are treated like any text tokens. When the model generates captions with sentence structures resembling those in the ground truth, even erroneous guesses of these statistics incur less loss than omitting them entirely.

**Case 2: Success**   The following is a success case where the finetuned `Idefics 2` is able to infer the time series mean accurately with a negligible error. We compare the captions generated by the pretrained and finetuned versions of the model against the ground-truth.

Interestingly, the issue of statistical overconfidence appears to be model-specific, as different models exhibit varying behaviors after finetuning. In this case, the finetuned `Idefics 2` was able to infer both the mean and the standard deviation with reasonable accuracy, when even the ground-truth caption did not explicitly include the standard deviation. This signals that some models benefit more from finetuning on our training data.

**Ground-Truth Caption**

The death rate in Papua New Guinea from 2017 to 2021 shows a relatively stable trend, fluctuating around a mean of **6.52** deaths per 1000 people, with a slight dip to 6.4 in 2020 and a peak of 6.68 in 2021. Compared to the global average, which steadily increased from 7.53 in 2017 to 8.57 in 2021, Papua New Guinea's death rate is significantly lower. While the global death rate saw a notable increase, particularly between 2020 and 2021, Papua New Guinea's rate remained relatively consistent, indicating a different trajectory than the global trend.

**Caption from Pretrained Model**

The yearly death rate in Papua New Guinea has been increasing since 2017. In 2017, the death rate was 6.51, and it has been increasing since then. In 2021, the death rate was 6.68, which is the highest value in the time series.

**Caption from Finetuned Model**

From 2017 to 2021, the death rate in Papua New Guinea, a lower-middle-income country, fluctuated slightly, ranging from a low of 6.40 in 2020 to a high of 6.68 in 2021. The average death rate during this period was **6.49**, with a standard deviation of **0.1**, indicating relatively stable numbers. Compared to the global average death rate, which decreased from 7.57 to 7.16 over the same period, Papua New Guinea's death rate was significantly lower and showed no clear downward trend.

Figure 18: Finetuned model successfully infers the mean and standard deviation with negligible error. Compare the **bolden** values.

## L    CATS-BENCH SAMPLES

In this section, we provide representative time series samples from CaTS-Bench across several domains. Samples include the numeric time series segment, rich JSON metadata, line plot image, and ground-truth caption.

**Time Series Segment**

```
7.0, 7.0, 7.0, 7.0, 7.0, 7.0, 7.0, 8.0, 25.5,
42.0, 163.33, 258.0, 214.5, 322.5, 354.75,
402.0, 182.33, 141.25, 69.25, 47.0, 12.5,
7.0, 7.0, 7.0, 7.0, 7.0, 7.0, 7.0, 7.0, 7.0,
7.0, 11.25, 84.0, 194.5, 338.75, 374.75,
427.25, 272.75, 332.67, 377.75, 232.33,
111.67, 113.25, 23.5, 10.0, 7.0, 7.0, 7.0,
7.0, 7.0, 7.0, 7.0, 7.0, 7.0, 7.0, 9.0,
72.25, 91.0, 54.5, 213.0, 299.25, 233.75,
259.0, 390.75, 367.75, 285.25, 207.75, 63.25,
9.5, 7.0, 7.0, 6.75, 6.75, 7.0, 7.0
```

**Metadata JSON**

```
{ "all-time maximum":  730.0, "all-time
average value until today":  127.62,
"all-time minimum":  0.0, "all-time standard
deviation until today": 175.79, "average
value in this time series": 105.78, "city":
"Visakhapatnam", "maximum value in this
time series":  427.25, "measure":  "SR
(W/mt2)", "minimum value in this time
series":  6.75, "sampling frequency":
"hourly", "standard deviation in this
time series": 133.47, "start_month":
"July", "start_year":  2017, "starting
time":  "2022-06-27 22:00:00", "state":
"Andhra Pradesh", "station_location":  "GVM
Corporation, Visakhapatnam " }
```

**Line Plot Image**

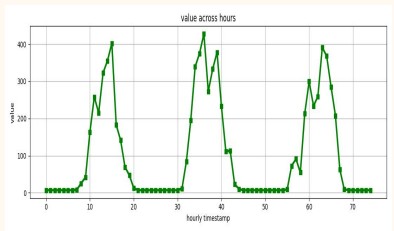

**Caption**

The hourly solar radiation (sr) data from Visakhapatnam, starting on June 27, 2022, exhibits a clear daily pattern of low values around 7 w/mt² during the night and early morning, sharply increasing to peaks during daylight hours, with a maximum value of 427.25 w/mt². compared to the city's all-time average of 127.62 w/mt², the average value in this time series is 105.78 w/mt². The data follows a consistent diurnal cycle, with repeated peaks during the day and low values at night, showing a stable daily pattern.

Figure 19: Sample 1 showing time series data, metadata, plot image, and reference caption.

**Time Series Segment**

[29, 33, 29, 35, 32, 35, 36, 34, 23, 34, 26, 25, 24, 30, 31, 39, 34, 48, 56, 40, 35, 37, 42, 55, 57, 41, 47, 53, 56, 50, 59, 45, 49, 44, 36]

**Metadata JSON**

{ "border": "US-Canada Border", "end date of the series": "2022-11-01", "general maximum in the history of this port": 80, "general mean in the history of this port": 30.85, "general minimum in the history of this port": 0, "general standard deviation in the history of this port": 13.74, "maximum in this specific series": 59, "mean of this specific series": 39.4, "means": "Trains", "minimum in this specific series": 23, "port": "Van Buren", "sampling frequency": "monthly", "standard deviation of this specific series": 10.16, "start date of the series": "2019-12-01", "state": "Maine" }

**Line Plot Image**

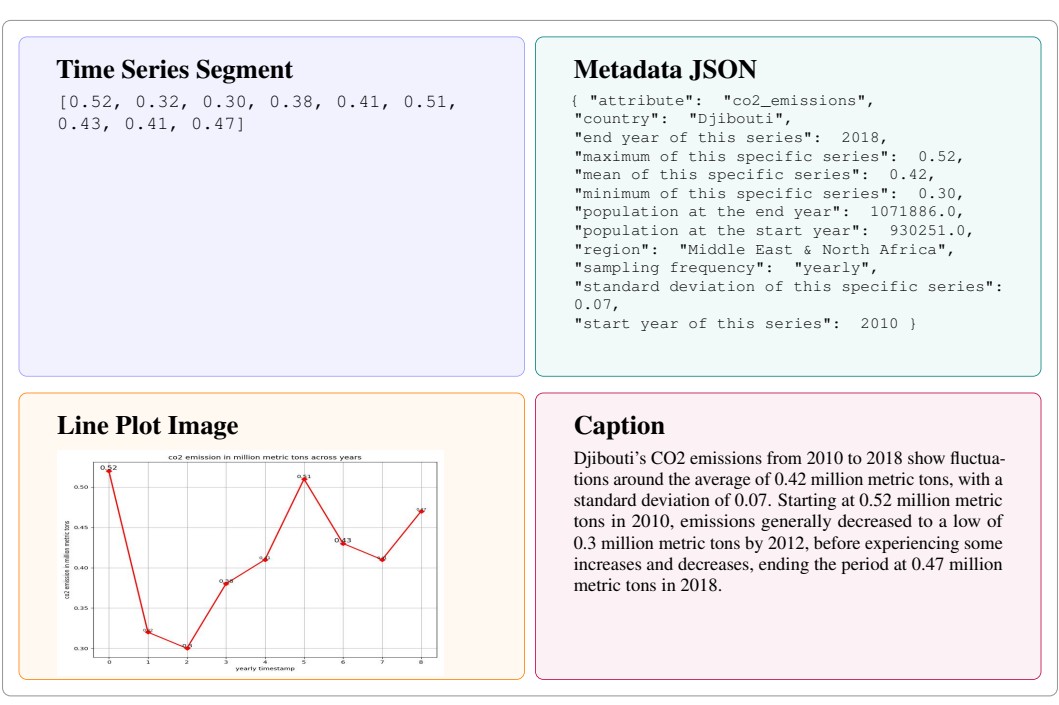

**Caption**

From December 2019 to December 2022, the number of monthly trains crossing the port of Van Buren generally increased, with an average of 39.4 trains, which is notably higher than the all-time mean of 30.85. The time series starts with values in the range of 23 to 36 trains until November 2020, and then experiences a significant upward shift, reaching a peak of 59 trains in October 2022. This increase suggests a moderate surge in cross-border traffic compared to historical trends.

Figure 20: Sample 2 showing time series data, metadata, plot image, and reference caption.

**Time Series Segment**

[0.52, 0.32, 0.30, 0.38, 0.41, 0.51, 0.43, 0.41, 0.47]

**Metadata JSON**

{ "attribute": "co2_emissions", "country": "Djibouti", "end year of this series": 2018, "maximum of this specific series": 0.52, "mean of this specific series": 0.42, "minimum of this specific series": 0.30, "population at the end year": 1071886.0, "population at the start year": 930251.0, "region": "Middle East & North Africa", "sampling frequency": "yearly", "standard deviation of this specific series": 0.07, "start year of this series": 2010 }

**Line Plot Image**

**Caption**

Djibouti's $CO_2$ emissions from 2010 to 2018 show fluctuations around the average of 0.42 million metric tons, with a standard deviation of 0.07. Starting at 0.52 million metric tons in 2010, emissions generally decreased to a low of 0.3 million metric tons by 2012, before experiencing some increases and decreases, ending the period at 0.47 million metric tons in 2018.

Figure 21: Sample 3 showing time series data, metadata, plot image, and reference caption.

**Time Series Segment**

[ 90.0, 90.0, 104.0, 104.0, 104.0, 104.0, 104.0, 104.0, 104.0, 110.57, 110.57, 110.57, 110.57, 110.57, 110.57, 121.29, 121.29, 121.29, ..., 268.86, 268.86, 257.29, 257.29, 257.29, 257.29, 257.29, 257.29, 266.14, 266.14, 266.14, 266.14, 266.14, 266.14, 266.14, 394.57, 394.57, 394.57, 394.57, 394.57, 394.57, 394.57, 411.57, 411.57, 411.57, 411.57, ..., 196.71, 196.71, 196.71, 143.43, 143.43, 143.43, 143.43, 143.43, 143.43, 143.43, 152.43 ]

**Metadata JSON**

{ "attribute": "cases", "country": "Thailand", "end date of this series": "2024-07-21", "historical maximum in this country": 26073.0, "historical mean in this country": 2877.0, "historical minimum in this country": 0.0, "historical standard deviation in this country": 5778.0, "income group": "Low & Middle Income", "maximum of this specific series": 465.14, "mean of this specific series": 240.7, "minimum of this specific series": 90.0, "population": 71697024, "region": "East Asia & Pacific", "sampling frequency": "daily", "standard deviation of this specific series": 106.03, "start date of this series": "2024-03-29" }

**Line Plot Image**

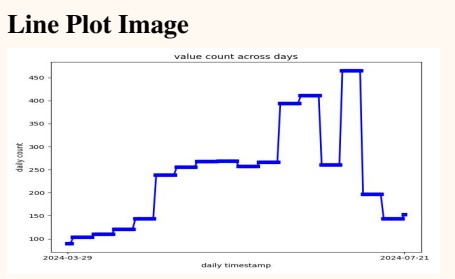

**Caption**

From March 29th to July 21st, 2024, the daily COVID-19 cases in Thailand fluctuated, starting at 90.0 and peaking at 465.14 around mid-June. This peak is significantly lower than the general maximum of 26073.0 daily cases observed in Thailand, indicating a substantial decrease in case numbers during this period. The time series shows an overall pattern of initial stability, followed by increases and decreases, with a final value of 152.43, suggesting a moderate decline towards the end of the observed period.

Figure 22: Sample 4 showing time series data, metadata, plot image, and reference caption.

**Time Series Segment**

[872817.62, 972716.24, 936508.43, 806979.15, 982322.24, 861894.77, 861941.25, 872469.03, 807798.73, 833517.19, 868191.05, 777207.3, 865924.2]

**Metadata JSON**

{ "attribute": "weekly_sales", "best week": "2010-12-24", "best week sales": 1309226.79, "end week of this series": "2012-08-31", "maximum of this specific series": 982322.24, "mean of this specific series": 870791.32, "mean sales": 893581.39, "minimum of this specific series": 777207.3, "sampling frequency": "weekly", "start week of this series": "2010-05-28", "worst week": "2010-12-31", "worst week sales": 635862.55 }

**Line Plot Image**

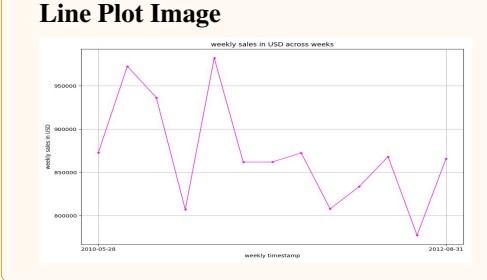

**Caption**

The provided sales data, spanning from May 2010 to August 2012, reveals fluctuations around a mean of $870,791.32, which is moderately below the overall store mean of $893,581.39. The lowest sales week within this period reached $777,207.3, while the highest peaked at $982,322.24, indicating a moderate level of volatility, but no extreme values compared to the store's best and worst weeks. The absence of a clear upward or downward trend suggests that sales remained relatively stable during this specific timeframe.

Figure 23: Sample 5 showing time series data, metadata, plot image, and reference caption.

# M EDITING AND REVIEW

## M.1 EDITS MADE IN THE CAPTIONS

During the editing process, captions were systematically refined to ensure accuracy, clarity, and consistency across the dataset. The following key rules were applied:

1. **Removal of external speculation:** Captions were restricted strictly to information verifiable from the metadata, time series, and plot, avoiding any causal claims or conjecture not grounded in the time series values or provided metadata.

2. **Variation in phrasing:** To reduce repetitiveness, sentence openings and phrases were varied rather than uniformly beginning and phrasing the same sentences.

3. **Pattern summarization:** When trends or unique structures (such as V-shaped or monotonic movements, etc.) were clearly visible, they were explicitly noted.

4. **Quantitative grounding:** Values such as maxima, minima, averages, and percentage changes were consistently included when relevant to ensure captions remained data-driven.

5. **Consistency with variation:** While maintaining factual accuracy and grounding in the data, captions were intentionally varied in structure and style to avoid monotony and ensure more natural, human-like phrasing across the dataset.

This systematic review process resulted in captions that were both faithful to the underlying data and stylistically coherent across the dataset.

## M.2 INTERFACE

In Figure 24, we provide a screenshot of the editing interface we used to edit the human-revisited test set.

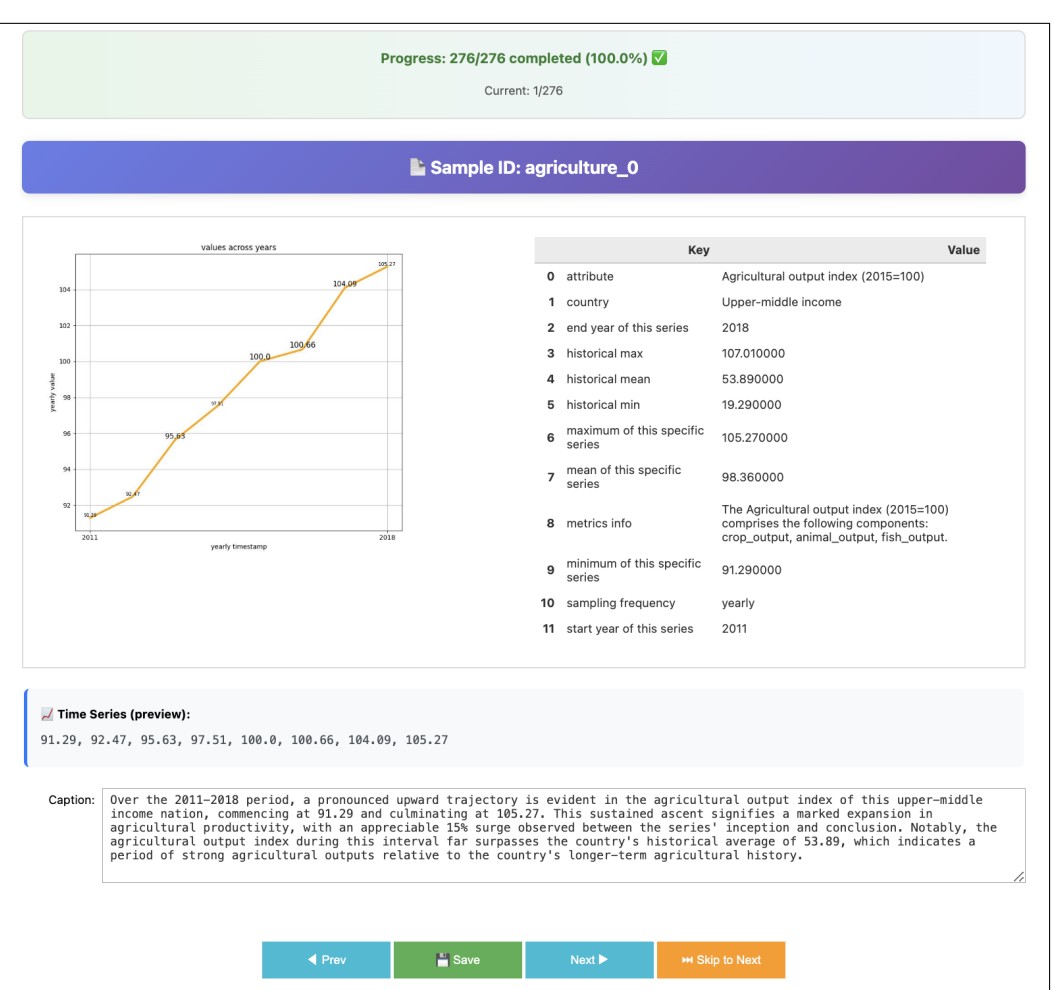

Figure 24: Interface used to edit and verify the captions.

# N TEMPLATE-BASED PROMPTS

In this section, we illustrate the prompts used in our sample generation pipeline, evaluation, paraphrasing, PAL, and distractor generation in Q&A. Angular brackets are used as placeholders for the actual values.

## N.1 GROUND-TRUTH CAPTION GENERATION PROMPT

The following is an example of a prompt for generating the ground-truth caption from the source dataset *Crime*.

```
Here is a time series about the number of <sampling frequency> crimes in
<town>, Los Angeles, from <start date> to <end date>:

<time series>

The all-time statistics of <town> until today are:
Mean: <general mean of this town>
Standard Deviation: <general standard deviation of this town>
Minimum: <general minimum of this town>
Maximum: <general maximum of this town>

And the statistics for this specific time series are:
Mean: <mean of this specific series>
Standard Deviation: <standard deviation of this specific series>
Minimum: <minimum of this specific series>
Maximum: <maximum of this specific series>

Describe this time series by focusing on trends and patterns. Discuss
concrete numbers you see and pay attention to the dates.

For numeric values, ensure consistency with the provided time series. If
making percentage comparisons, round to the nearest whole number. Report
the dates when things happened.

Use the statistics I provided you for comparing this example to the
normalcy.
Do not add any extra information beyond what is given.
Highlight significant spikes, dips, or patterns.

You don't have to explicitly report the numeric values of general
statistics; you just use them for reference.
Compare the trends in this time series to global or regional norms,
explaining whether they are higher, lower, or follow expected seasonal
patterns.
When making comparisons, clearly state whether differences are minor,
moderate, or significant.

Use descriptive language to create engaging, natural-sounding text.
Avoid repetitive phrasing and overused expressions.

Answer in a single paragraph of four sentences at most, without bullet
points or any formatting.
```

## N.2 BASELINE CAPTION GENERATION PROMPT

When evaluating the baselines on our benchmark, we provide limited metadata, excluding the precomputed statistics of the time series, as the models are expected to infer them on their own. An example of the prompt is the following.

```
Here is a time series about the number of <sampling frequency> crimes in
<town>, Los Angeles, from <start date> to <end date>:
```

```
<time series>

Describe this time series by focusing on trends and patterns. Discuss
concrete numbers you see and pay attention to the dates. For numeric
values, ensure consistency with the provided time series. If making
percentage comparisons, round to the nearest whole number. Report the
dates when things happened.

Compare the trends in this time series to global or regional norms,
explaining whether they are higher, lower, or follow expected seasonal
patterns.

When making comparisons, clearly state whether differences are minor,
moderate, or significant.

Use descriptive language to create engaging, natural-sounding text.
Avoid repetitive phrasing and overused expressions.

Answer in a single paragraph of four sentences at most, without bullet
points or any formatting.
```

### N.3   CAPTION PARAPHRASING PROMPT

To rephrase a caption into a different linguistic style while preserving semantic and numeric information, we feed the following prompt into a paraphraser model of choice.

```
You are a helpful assistant. Your task is to rephrase the following
paragraph that describes a time series. You MUST strictly follow these
rules:

Preserve all factual information: All numeric values, statistics (min,
max, mean, etc.), trends ('increased', 'peaked'), comparisons ('higher
than'), and dates must remain exactly the same.

Change the style completely: Use different sentence structures,
synonyms, and grammatical constructions. Alter the tone (e.g., make it
more formal or more conversational). Do not use the same phrasing as the
original.

Output only the rephrased paragraph, with no additional explanation.

Here is the paragraph to rephrase:
<caption>
```

### N.4   PAL PROMPT

Here we reproduce the full prompt used for the program-aided context:

```
### Task
<caption_prompt>

### Instructions for the assistant
1. You are an expert coding assistant; think through the task
**step-by-step**.
2. Write **Python 3.12** code (inside one ```python``` block) that
computes the final answer.
   * Use only the Python Standard Library (e.g., you may use the `math`,
`statistics` libraries).
   * Wrap everything in a `solve()` function that will be invoked to
produce the final caption.
   * The code **must produce the caption string itself**. Any numeric
values can be computed
```

```
      in Python and formatted into the caption string. Make sure to use
any values you compute
      in the resulting caption string.
3. The `solve()` function you write will be invoked to produce the final
caption.

### Output format (exactly; no extra text, explanations, or formatting)
```python
# code that defines solve() and any desired strings
solve()
```
```

The full TSC prompt from N.2 is injected as the `caption_prompt` string.

### N.5  SEMANTIC PERTURBATION PROMPT

To perturb a caption so that its semantic meaning is altered while keeping numbers intact, we feed the following prompt into `Gemini 2.0 Flash`.

```
Your task is to minimally modify a time series description so that its
meaning is altered but the numbers are maintained.

For example, you can switch "increase" with "decrease", "upward" to
"downward", or something more sophisticated. Keep the description
structurally identical to the original text; you don't have to alter too
much information. Altering anywhere between 1 and 3 parts is enough. Do
not edit the numbers.

Here's the description to modify:
<caption>

Give your answer in a paragraph of text as the given description,
without any explanation or formatting.
```

### N.6  NUMERIC PERTURBATION PROMPT

To perturb a caption so that its numbers are altered while its semantic information is preserved, we feed the following prompt into `Gemini 2.0 Flash`.

```
Your task is to slightly modify the numbers in a time series description
so that its semantics remain the same, but the numbers are slightly
altered.

For example, you can replace "12" with "12.2", "45%" with "46%". Keep
the description structurally and semantically identical to the original
text; you don't have to alter all numbers but anywhere between 1 to 3
times is enough. Make sure that the altered number still makes sense and
fits the scale of the phenomenon.

Here's the description to modify:

<caption>

Give your answer in a paragraph of text as the given description,
without any explanation and formatting.
```

## O  HUMAN BASELINE

To establish a human performance baseline, we invited university students to voluntarily complete all four Q&A tasks. These tasks span a range of reasoning types, including fine-grained statistical

comparisons, semantic interpretation, and multimodal alignment. Participants were recruited through academic networks and completed the tasks without the aid of external tools, ensuring a fair comparison with models operating under similar conditions. Participation was entirely voluntary, with no compensation, and individuals could withdraw at any time. Below, we present the instructions given to the volunteers for their participation.

```
Participant Information and Consent Form for Time Series QA
Questionnaire

Thank you for considering participation in our study!

This questionnaire is part of a research project evaluating human
performance on time series understanding tasks. Your responses
will help us establish a baseline for comparing human performance
to that of current language models. You will be given a Google
Form consisting of 10 to 14 multiple-choice questions of the same
type, and you should not use any external tools.

Please read the following information carefully before continuing:

Voluntary Participation: Your participation is entirely
voluntary. You may choose not to participate or to withdraw at
any time without any consequences.

Duration: The questionnaire is brief and is estimated to take
between 3 and 6 minutes to complete.

Anonymity & Data Use: No personal information will be collected
or stored. Your answers will remain anonymous and will be used
solely for research purposes, such as evaluating and reporting
model performance in academic publications.

No Compensation: There is no monetary or material compensation
for participating in this study.

Confidentiality: All collected data will be handled securely.
Only aggregated and anonymized results will be published.

By proceeding, you confirm that you understand the above terms
and agree to participate in this research study.

Thank you for your collaboration and contribution to our research!

Date: ____________________        Signature: ____________________
```

