# OpenReview forum: "CaTS-Bench: Can Language Models Describe Numeric Time Series?"
_ICLR.cc/2026/Conference — ICLR 2026 Conference Withdrawn Submission_

### Official Review · Reviewer_ghrj · 2025-10-29

**Soundness:** 3
**Presentation:** 3
**Contribution:** 2
**Rating:** 6
**Confidence:** 4

**Summary:**

This work proposed a large scale time series captioning benchmark together with some multiple choice questions on time series and text alignments.

**Strengths:**

Overall, this work makes a valuable attempt to define the task of time series captioning and introduces a relatively large dataset with a degree of human verification. It could represent an interesting contribution for the community, particularly regarding time series-text alignment module pretraining. That said, I am uncertain whether this task is a clear subset of chart understanding / OCR tasks, which have been actively explored. However, it seems the authors intentionally focused their discussion on time series.


Overall, the work appears reasonable and engaging for the following reasons:
1. Multiple strategies are employed to address concerns about semi-synthetic caption generation using LLMs, including manual verification and a human detectability study.
2. The question answering dataset offers some interesting alignment tasks between time series and text.

**Weaknesses:**

1. The manuscript could benefit from deeper analysis of the reported results. It would be helpful to clarify what specific weaknesses of current VLMs/LLMs are revealed through this task. In Table 3, the performance on semi-synthetic captions appears higher than that on human-verified ones. Any insight into why this occurs would strengthen the discussion.

2. Since the Q&A component is repurposed from the captions, and the captions themselves are not perfect, it raises some concern about the robustness of the resulting subset. It would be valuable to evaluate its stability through some form of repeated sampling or uncertainty estimation.

3. In Section 3.5.2, it might be clearer to report standard metrics such as F1, Precision, and Recall separately for numerical scores. Using a weighted combination as a final metric could introduce bias, as the overall score may vary depending on the chosen value.

4. While the authors mention the scalability of the pipeline, the manuscript could better demonstrate the practical value of this scalability. It would therefore be useful to include experiments or analyses showing how model performance scales with dataset size and whether it eventually plateaus or continues to improve.

**Questions:**

It is expected that a program-aided LLM could surpass its base model on computation heavy tasks, given its access to code execution. Could a more capable LLM agent further improve performance under this framework?

---

### Official Review · Reviewer_5khQ · 2025-10-31

**Soundness:** 2
**Presentation:** 2
**Contribution:** 2
**Rating:** 2
**Confidence:** 3

**Summary:**

The authors propose a benchmark for a task they claim to be “Context-Aware Time Series Captioning” where each sample includes a time series, metadata about the time series, a line-chart image of the time series, and a “caption”. They propose a train-test split for this benchmark and a scalable pipeline to generate reference captions with an LLM, a fraction of which are verified by humans. They also evaluate LLMs/VLMs for a few other problems apart from captioning, such as TS matching, caption matching, plot matching and TS comparison.

**Strengths:**

- The one task from this paper that makes sense to me is the caption matching task and plot matching tasks, that are designed to understand the multimodal alignment of LLMs.

- Another idea discussed in the paper is about LLMs predicting “trend descriptions” from time series. This task makes sense as a standalone task to me.

- I like the “Visual Modality Ablation” that the authors present in Sec 4.3. Given the benchmark and evaluations, this ablation is interesting and makes sense.

- A finding that “A consistent weakness lies in multimodal grounding: models largely ignore visual inputs, with plot matching emerging as the most difficult task.” is interesting and would be useful to the community.

**Weaknesses:**

- The paper is very confusing, and poorly presented; I was unable to understand even the problem setting despite multiple re-reads; I was even further unable to understand the contribution of the authors. An example of a very confusing paragraph would be the first paragraph of Sec 3.4, which I still have a hard time understanding.

- The paper in the first paragraph of its introduction claims that the need to “interpret” time series data motivates a task that requires converting numeric statistics and metadata to a “caption”.
However this is not well-motivated; I do not see any instance of such a task being useful in the real-world. In real-world cases, the structured metadata is more useful than an unstructured caption.
Despite re-reading the paper several times, I am unable to understand what value the “caption” adds over the statistics that you can compute with tools like pandas (like mean, median, outliers).
  - I.e. What information is added by the caption that is not available (metadata) or can be computed (e.g. mean statistics) in structured form?
  - This question has plagued me throughout my reading of the paper; and I hence do not see any need for this problem setting. Even if such a problem setting is realistic, I don’t see why we need LLMs for this problem setting when one can simply use the metadata, compute statistics etc. and make a “caption”.
  - I see the lack of a good motivation for the problem setting as a weakness.

- I feel contribution (1) and (2) given on Page 2 overlap significantly in terms of what’s mentioned about human verification. I”m still confused about where exactly in the pipeline human verifications have been done, as the contributions are discussed in the paper in a very confusing way.
  - I suggest the authors expand Figure 2 to mention all instances of human verification that the pipeline uses (the figure currently has no mention of human verification)
  - This is related to my point on the paper being confusing to understand, which is a major weakness.

- How exactly is the generated caption different from the final prompt? (In Figure 2). This is central to the paper but is not clear from the figure, the paper or the appendix.

- The authors state “a ground truth caption produced by querying an oracle LLM (Gemini 2.0 Flash) with a structured prompt that includes: (i) the serialized numeric values of the cropped segment and (ii) metadata enriched with numericly grounded information, including both the historical and sample-specific mean, standard deviation, minimum, and maximum.”
   - Isn’t this the task that LLMs benchmarked on this benchmark need to solve? If the ground truth itself is obtained from an LLM, despite any human modifications, it means that  the task is not that difficult for LLMs in the first place.
   - I read the prompt mentioned in Appendix N.1: I wonder what statistics are expected from the LLM which are not derivable directly from the data using statistics. I just don't see an LLM being required for the entirety of this task.
   - The problem setting does not make sense here; since this is central, I count this as a weakness. A few of my other points are linked to this weakness but I split them into different points as they are questions that are re-iterated in different sections of the paper.

- The authors state “The oracle receives full contextual metadata (not available at evaluation time) and is instructed not to include any external knowledge, ensuring captions remain factual and context-grounded”
   - So if I’m correct, when models are trained for this task, the only difference between training and test time is that the statistics of a time series window are not given at test time. However these statistics can all be computed in the fraction of a second using pandas. Why not provide them? In this case, is the goal of the LLM just to learn to predict these statistics (which doesn’t make sense)?
 - What do the authors check for in the generated caption by the oracle? What information does it have that is not present in metadata?

- The authors state “During evaluation, each model is presented with a standardized multi-part prompt that combines four elements:”
    - Exactly my point, if all these statistics are provided to the LLM, what do the authors expect the LLM to predict, that is not already available in the input?

- In Sec 3.3, the authors state  “a fixed-format instruction template containing the directive for caption generation”
	- The prompt says “"Compare the trends in this time series to global or regional norms, explaining whether they are higher, lower, or follow expected seasonal patterns."
    - Without the metadata of the global time series given as input, what is the model expected to predict?”


- Following all the above questions, if the authors expect the model to predict something that is not already available in the metadata or input prompt, why not ask the model to directly predict that quantity (e.g. trend descriptor)? This makes the task well-defined, the evaluation more direct, and avoids any ambiguity in the captions.

- Both the metrics for numeric fidelity evaluate for specific quantities mentioned in the caption; instead of this why not prompt the LLM to just predict what needs to be evaluated for?

**Note**: I re-read the paper multiple times and spent several hours in this review trying to understand the paper better. As the problem setting and central contributions of this paper are not well-motivated, I propose to reject it. I hope the authors find enough information in my review to understand the reason behind my decision.

**Questions:**

I list here only questions that I need further clarification on, which I don't count in the score of the paper. The score of the paper only takes into account the major points, all of which are listed in the weaknesses. I carefully split them so it is clear what factors into my score.

- Suggestion/Question to the authors: Do you give in-context examples to LLMs when evaluating them zero-shot? This seems like a task that might be poorly specified with a single prompt.
Just giving an example of what you would want the LLM to output, might improve performance

- Figure 3: Which models are proprietary, pretrained and finetuned? It is not in the legend of the figure.


- The authors state “Our comprehensive experiments on leading VLMs reveal that, in both zero-shot and finetuned settings, models can produce fluent text but fail to reliably capture quantitative details without specialized adaptation.”
What is “specialized adaptation” here beyond finetuning?

- In the “Time Series Comparison” task, are the jargons (e.g. volatility etc.) defined when the LLM is prompted?

---

### Official Review · Reviewer_LHBb · 2025-10-31

**Soundness:** 3
**Presentation:** 3
**Contribution:** 2
**Rating:** 4
**Confidence:** 3

**Summary:**

The paper presents a new dataset for evaluating time series reasoning. The authors take several existing time series datasets and augment them by using a language model to write captions. Extensive experiments are conducted to evaluate the quality of the descriptions.

**Strengths:**

- It seems substantial time and effort went into creating the "human-revisited" subset.
- Experiments, including ablations, error analysis, annotator agreement, comparison between human and LLM captions are all very good and impressive.
- I appreciate the analysis of not just VLMs but also text-only models. Several similar papers in the past have ignored one modality or the other.

**Weaknesses:**

There is, of course, the weakness that the captions are LLM-generated rather than written by human annotators or found "in the wild" with the time series themselves. That being said, the authors conduct many good experiments to alleviate the obvious concerns.

**Questions:**

- Why do we need "metadata and visual representations" as the abstract suggests? I don't understand why this is important, and the abstract could be improved by explaining why this is the case. Surely any time series dataset could be trivially plotted to get the visual representation?
- "Evaluation resources fail to reflect the complexity of real-world temporal signals, leaving model improvements unguided by the demands of true data-driven applications". Can you prove that this is the case for the other benchmarks you mention? Just saying "they're not complex enough" or "not real enough" isn't sufficient evidence to explain why the world needs your benchmark. Maybe it would be useful to show examples from these other datasets and explain exactly why they fall short compared to what you're offering?
- What exactly makes the pipeline "scalable"? It seems that human supervision of the pipeline is still necessary to ensure quality.

---

### Official Review · Reviewer_PNih · 2025-11-02

**Soundness:** 2
**Presentation:** 3
**Contribution:** 2
**Rating:** 2
**Confidence:** 5

**Summary:**

The paper proposes CaTS-Bench, a large-scale multimodal (time series and text) benchmark for real-world time series captioning and question-answering benchmark. The benchmark is based on multiple real-world datasets, and the captions are generated automatically using a large language model supplied with meta-data about the time series in question. The authors design 4 types of multiple choice questions to evaluate capabilities of large language models in time series matching, caption grounding, visual reasoning, and comparative analysis.

**Strengths:**

The paper addresses an important and understudied problem, and provides a large time series captioning dataset, along with a flexible pipeline to generate similar datasets. I really like the design of the distractors.

**Weaknesses:**

1. **Captions are not reflective of real-world use:** It is not immediately clear, how the time series captions are useful and reflective of real world use. There are very domains where time series captions are relevant, for example clinical interpretations (used in [1]), financial reporting (e.g., earnings calls, analyst reports), weather reporting etc. But none of these practical settings were considered. Instead, the time series captioning problem was reduced to "describing" a time series. Consider the example in Figure 22: Without access to the metadata, and just by looking at the line plot image, there is no way _any_ model will be able to reason about the maximum within this figure (i.e. 465.14), or the overall maximum (26073.0). My fundamental issue with the paper is that, time series captioning, as described in the paper has no practical use. Moreover, like the authors mention "time series captioning lacks inherent ground truth at the level of a single canonical description". This further reduces the utility of the proposed benchmark.

2. **LLM-based caption generation is unreliable**: The authors use LLMs to generate time series captions. However, LLMs are unreliable, and their performance in understanding time seres data is questionable at best (as shown in prior work [1, 2], and also in this study). How can we then rely on the generated dataset? The authors might argue that the LLM has access to metadata, which the LLM can condition on, to generate the caption. This metadata comprises of information already in the time series (e.g., max, mean etc.), and extrinsic information (e.g., overall maximum). Since during inference, the LLM does not have this metadata, it cannot realistically infer extrinsic information. On the other hand, one can use the LLM for intrinsic information, but it is an overkill.

3. **Large-scale captioning dataset:** The large-scale nature of this dataset is a major contribution, however, only a small subset of captions were verified by humans. The authors claim that the verified dataset had some error modes (see Question #1). Thus, the quality of the larger dataset is questionable.

4. **Lack of discussion and comparison with prior work:** The paper lacks discussion of multiple studies in the intersection of time series and language, for example JoLT [1] which solves the problem of time series captioning in the context of clinical text, and TimeSeriesExam, which proposed a scalable benchmark to evaluate LLM's understanding of time series data, and relevant background on LLMs and time series [3]

> whether captions generated by the oracle model (Gemini 2.0 Flash) are factual, unbiased, and linguistically diverse.
5. How about realism? Are the captions similar to real-world captions for the same domains? What notion of reasoning are the Q&A tasks evaluating? The tasks are not realistic-- I am not aware of anyone solving these tasks. Figure 3 for example compares models on mean matching. Why would I use an LLM to do that?

> We introduce a suite of multiple-choice Q&A tasks designed to probe different reasoning skills in time series understanding.
6. **Time series captioning and reasoning are under-specified problems**: Both time series captioning and reasoning are under-specified problems. The authors provide no formal or working definition of reasoning. There have been prior studies which define the notion of reasoning [2, 4] to some extent, and I would encourage the authors to consider and build upon their formalisms.

7. In section 3.5, I recommend that you describe the type of questions before mentioning the filtering process.

8. **LLMs prefer text generated by them:** As the authors have shown, LLMs are well known to favor content generated by them, and Table 3 demonstrates this clearly. While the authors have conducted a study to refute this, I wonder if the trends would hold if another LLM was used to generate the captions.

### References
1. Cai, Yifu, et al. "Jolt: Jointly learned representations of language and time-series." Deep Generative Models for Health Workshop NeurIPS 2023. 2023.
2. Cai, Yifu, et al. "Timeseriesexam: A time series understanding exam." arXiv preprint arXiv:2410.14752 (2024).
3. Zhang, Xiyuan, et al. "Large language models for time series: A survey." arXiv preprint arXiv:2402.01801 (2024).
4. Potosnak, Willa, et al. "Investigating compositional reasoning in time series foundation models." arXiv preprint arXiv:2502.06037 (2025).

**Questions:**

> These captions were first sampled from multiple LLM candidates (Gemini 2.0 Flash, GPT-4o, Gemma 27B, and Llama 90B) using the above data pipeline, and then carefully refined by the authors to eliminate factual errors, speculative statements, and redundant phrasing.


1. What proportion of questions had to be modified? What were the failure modes? What is the guarantee that the rest of the unverified questions did not have the same error modes?

2. Human Detectability Study: What instructions were provided to the judges? What was the experimental setting?

3. Linguistic style experiment: Why did you paraphrase the caption, why not use other LLMs to generate novel captions in the first place?

---

### Note · Authors · 2025-11-24

**Comment:**

Dear Area Chairs and Reviewers,

We would like to thank you for the time, effort, and thoughtful feedback provided in reviewing our submission. We sincerely appreciate the thorough evaluations, constructive critiques, and detailed questions.

After carefully considering the reviews, we have decided to withdraw the paper. The comments highlighted important issues in our current formulation, and we believe that addressing them properly will require substantial redesign and additional experimentation.

We are grateful for the reviewers’ insights, which will significantly guide the next iteration of this work. Thank you again for your effort and for helping us improve our research.

**Withdrawal Confirmation:**

I have read and agree with the venue's withdrawal policy on behalf of myself and my co-authors.